# Transcriptomic changes including p53 dysregulation prime DNMT3A mutant cells for transformation

Erin M Lawrence [1,2,7], Amali Cooray [1,2,7], Andrew J Kueh[1,2,3,4], Martin Pal [1,2,5], Lin Tai [1,3], Alexandra L Garnham[1,2], Connie S N Li-Wai-Suen [1,2], Hannah Vanyai[1,2], Quentin Gouil [1,2,3,4], James Lancaster[1,2], Sylvie Callegari [1,2], Lauren Whelan[1,2], Elizabeth Lieschke [1,2,6], Annabella Thomas [1,2], Andreas Strasser [1,2], Yang Liao [3,4], Wei Shi[3,4], Andrew H Wei[1,2] & Marco J Herold [1,2,3,4 ✉]

## Abstract

*DNMT3A* mutations are prevalent in haematologic malignancies. In our mouse model the murine homologue (R878H) of the human 'hotspot' R882H mutation is introduced into the mouse *Dnmt3a* locus. This results in globally reduced DNA methylation in all tissues. Mice with heterozygous R878H *DNMT3A* mutations develop γ-radiation induced thymic lymphoma more rapidly than control mice, suggesting a vulnerability to stress stimuli in *Dnmt3a*$^{R878H/+}$ cells. In competitive transplantations, *Dnmt3a*$^{R878H/+}$ Lin⁻Sca-1⁺Kit⁺ (LSK) haematopoietic stem/progenitor cells (HSPCs) have a competitive advantage over WT HSPCs, indicating a self-renewal phenotype at the expense of differentiation. RNA sequencing of *Dnmt3a*$^{R878H/+}$ LSKs exposed to low dose γ-radiation shows downregulation of the p53 pathway compared to γ-irradiated WT LSKs. Accordingly, reduced PUMA expression is observed by flow cytometry in the bone marrow of γ-irradiated *Dnmt3a*$^{R878H/+}$ mice due to impaired p53 signalling. These findings provide new insights into how *DNMT3A* mutations cause subtle changes in the transcriptome of LSK cells which contribute to their increased self-renewal and propensity for malignant transformation.

**Keywords** DNMT3A; Epigenetics; DNA Methylation; Haematological Cancers; Genetic Engineering
**Subject Categories** Cancer; Chromatin, Transcription & Genomics; Signal Transduction

## Introduction

Mutations in epigenetic modifiers are common tumour initiating mutations in haematologic cancers. One of the most frequent mutations present in haematologic malignancies occurs in the gene encoding DNA methyltransferase 3 alpha (*DNMT3A*) (Fong et al, 2014; Venney et al, 2021; Yang et al, 2015b). While a mutation in *DNMT3A* alone is not sufficient to cause malignancy (Lu et al, 2016; Xu et al, 2014), it is a potent driver of clonal haematopoiesis (CH) (Scheller et al, 2021). It is widely understood that *DNMT3A* is essential for maintaining the balance between haematopoietic stem cell (HSC) self-renewal and differentiation (Challen et al, 2011). Genetic loss of *DNMT3A* causes an almost indefinite HSC self-renewal phenotype (Challen et al, 2011; Challen et al, 2014; Jeong et al, 2018), which may explain its activity as a common driver of CH in humans. However, not all *DNMT3A* mutations are functionally equal. In human haematologic cancers there is a considerable bias towards mutations at arginine 882 (*DNMT3A*$^{R882}$) (Lu et al, 2019; Xu et al, 2014), most commonly an R882H mutation, indicating that this mutation confers a possible competitive advantage to affected cells. Indeed, *DNMT3A*$^{R882H}$ mutations are associated with poor patient outcomes in acute myeloid leukaemia (AML), highlighting the increased pathogenicity of the *DNMT3A*$^{R882H}$ mutation in patients and a critical need for targeted therapies (Guryanova et al, 2016b; Ley et al, 2010).

The effect of the *DNMT3A*$^{R882H}$ mutation in a pre-leukaemic setting is not well understood. The murine homologue of the R882H mutation is R878H, and *Dnmt3a*$^{R878H}$ mouse models (Dai et al, 2017; Liao et al, 2022) have previously been generated to examine the impact of this mutation in the initiation of CH and AML. To extend this work of others (Smith et al, 2021; Wang et al, 2024), we used CRISPR/Cas9 gene editing to generate a *Dnmt3a*$^{R878H/+}$ mouse model. Unlike previous Cre inducible *Dnmt3a*$^{R878H}$ models, this allowed us to investigate the impact of *Dnmt3a*$^{R878H}$ mutations with minimal intervention, avoiding the risk that Cre mediated gene (Loberg et al, 2019) recombination itself could impact the phenotype. Using this model, we explored phenotypic and transcriptomic changes in pre-leukaemic cells that caused the accumulation of clones in the blood, and investigated why these cells are at an increased risk of malignant transformation.

*DNMT3A* mutations may also have hitherto unrecognised non-canonical functions in HSCs. *TP53* in humans, or *Trp53* in mice

¹Walter and Eliza Hall Institute of Medical Research, Parkville, Victoria, Australia. ²Department of Medical Biology, University of Melbourne, Parkville, Victoria, Australia. ³Olivia Newton John Cancer Research Institute, Heidelberg, Australia. ⁴School of Cancer Medicine, La Trobe University, Bundoora, Victoria, Australia. ⁵School of Dentistry and Medical Sciences, Charles Sturt University, Wagga Wagga, NSW, Australia. ⁶Oncogene Biology Laboratory, Francis Crick Institute, London, UK. ⁷These authors contributed equally: Erin M Lawrence, Amali Cooray. ✉E-mail: Marco.Herold@onjcri.org.au

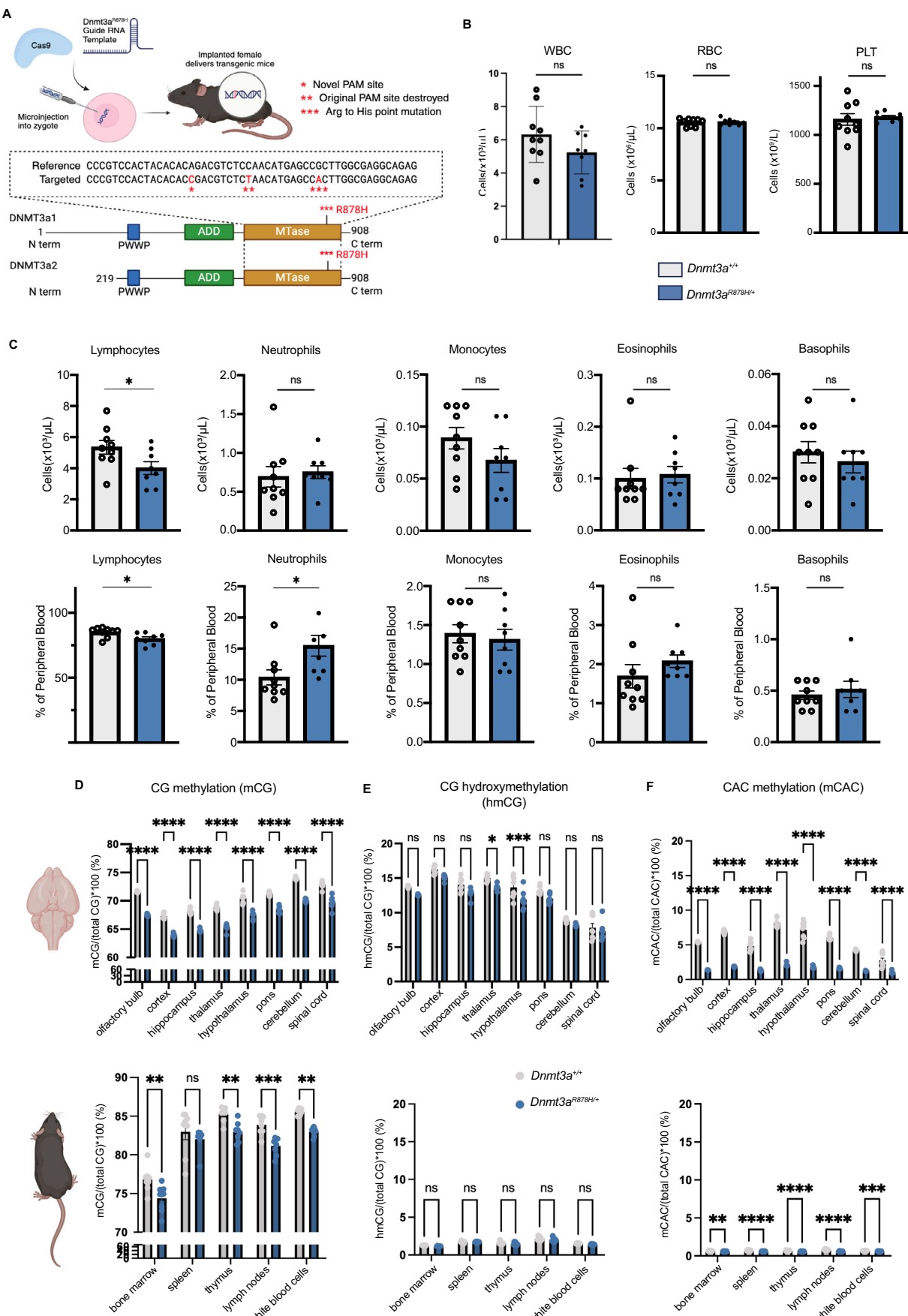

**Figure 1. *Dnmt3a^{R878H/+}* mutant mice have reduced DNA methylation but normal haematopoietic cell populations.**

(A) Schematic depicting the design of the *Dnmt3a^{R878H}* mouse model. Three point mutations were introduced using CRISPR/Cas9 gene editing technology to create a novel PAM allowing the mutant allele to be specifically targeted (*), destroy the original PAM site to prevent Cas9 targeting the correct allele (**), and introduce the *Dnmt3a^{R878H}* mutation into exon 23 of the endogenous *Dnmt3a* gene (***). (B) White blood cell (WBC), red blood cell (RBC) and platelet (PLT) counts from 8- to 10-week-old *Dnmt3a^{R878H/+}* and WT mice, collected by retro-orbital bleeds. Cell counts were determined by using an ADVIA analyser. (C) Mature blood cells in 8–10-week-old *Dnmt3a^{R878H/+}* and WT mice presented by both number and by percentage. (D–F) Different methylation readouts: mCG (D), hmCG (E), and mCAC (F), presented as a percentage of total CG in the indicated brain regions and other tissues from *Dnmt3a^{R878H/+}* and WT mice. %mCG is presented with a break in the y-axis to better visualise differences between genotypes. Data information: (B, C) Error bars are the mean (±SEM) of $n = 8$ *Dnmt3a^{R878H/+}* and $n = 9$ *Dnmt3a^{+/+}* independent biological repeats. Statistical significance was assessed using Prism 10 software by t tests. ns, not significant; $p > 0.05$, *$p < 0.05$. (D–F) Error bars are the mean (±SEM) of $n = 8$ *Dnmt3a^{R878H/+}* and $n = 8$ *Dnmt3a^{+/+}* independent biological repeats. Statistical significance was assessed using Prism 10 software by 2-way ANOVA with Šídák's multiple comparisons test. ns, not significant; $p > 0.05$, *$p < 0.0332$, **$p < 0.0021$, ***$p < 0.0002$, ****$p < 0.0001$.

(generally referred to as *p53* throughout this text) is a key tumour suppressor gene. There is emerging literature associating *p53* dysfunction with mutant *DNMT3A*, although it is not clear if this contributes to any disease phenotypes (Tuval et al, 2022; Wang et al, 2005). By exploring how changes in the epigenome and transcriptome may contribute to the initiation of malignancy in *DNMT3A* mutant cells, we identify disease inducing mechanisms that are specific to *DNMT3A* mutant cells. We anticipate that this will aid the development of therapies that will improve the survival of patients with *DNMT3A* mutant driven pathologies.

## Results

### Generation of a *Dnmt3a^{R878H/+}* mouse model using CRISPR/Cas9 gene editing technology

The *Dnmt3a^{R878H/+}* mouse model was created using CRISPR/Cas9 gene editing technology (Fig. 1A). Male *Dnmt3a^{R878H/+}* mice were inter-crossed with female C57BL/6 mice to generate heterozygous *Dnmt3a^{R878H/+}* mice. As previously reported (Smith et al, 2021), female *Dnmt3a^{R878H/+}* mice had birthing difficulties and were not used for breeding.

To determine the impact of the *Dnmt3a^{R878H/+}* mutation on haematopoiesis, peripheral blood analysis was performed on *Dnmt3a^{R878H/+}* mice and wild-type (WT) littermates at 8–10 weeks of age. There were no significant differences in white blood cell (WBC), red blood cell (RBC) or platelet counts (Fig. 1B). However, *Dnmt3a^{R878H/+}* mice had fewer lymphocytes by both percentage and absolute number, and this coincided with a proportional increase in neutrophils by percentage. Other blood cell populations appeared unaffected by the mutation (Fig. 1C). Further analysis of the blood and spleen by flow cytometry revealed no additional abnormalities in the *Dnmt3a^{R878H/+}* mice (Fig. EV1A,B).

### *Dnmt3a^{R878H/+}* mice exhibit reduced DNA methylation across several organs

The consequence of the *Dnmt3a^{R878H/+}* mutation on global DNA methylation was assessed by low-coverage direct DNA sequencing (Faulk, 2023). The epigenetic process of DNA methylation involves the transfer of a methyl group to carbon 5 of a cytosine, forming 5-methylcytosine (5mC). The addition of a hydroxyl group to 5mC generates 5-hydroxy-methyl-cytosince (5hmC), which may be an intermediate stage in DNA demethylation that can contribute to regulation of gene expression (He et al, 2021; Moore et al, 2013). In

mammals, CG methylation (mCG) which occurs at cytosines preceding guanines, also called CpG sites, is the most common type. Non-CpG methylation, such as in the methylated cytosine-adenosine (mCA) di-nucleotide context, occurs less frequently, and primarily in the brain (de Mendoza et al, 2021) and in pluripotent stem cells (Mao et al, 2020; Tan et al, 2019). Several lymphoid tissues, as well as the brain, which was further divided into regions, were collected from 8 *Dnmt3a^{R878H/+}* mice and 8 WT littermates. The genomic DNA was extracted, and an Oxford Nanopore Technologies' nanopore sequencer was used to perform sequencing of extracted DNA from each tissue at low coverage (0.01–0.05X) to measure the global levels of cytosine modifications on autosomes. Apart from the spleen, every tissue examined had a consistent reduction of mCG in *Dnmt3a^{R878H/+}* mice compared to WT littermates (Fig. 1D). The reduction in methylation was strongest in the brain regions, and more subtle in other tissues. This global hypomethylation phenotype is consistent with previous reports (Beard et al, 2023; Jeong et al, 2018; Li et al, 2023). Hydroxymethylation of CpG sites (hmCG) was found to be significantly reduced in the thalamus and hypothalamus of *Dnmt3a^{R878H/+}* mice, while there was no significant reduction in any lymphoid tissues examined, which have much lower prevalence of hmCG compared to the brain (Kinde et al, 2015) (Fig. 1E). Notably, CAC methylation (mCAC), a highly specific DNMT3A-catalysed modification (Mao et al, 2020), was substantially reduced in *Dnmt3a^{R878H/+}* mice across all tissues tested (Fig. 1F).

### *Dnmt3a^{R878H/+}* haematopoietic stem and progenitor cells are primed for γ-irradiation induced thymic T cell lymphomagenesis

A *DNMT3A* mutation may be among the first to arise in the progression from normal to malignant haematopoiesis (Mayle et al, 2015), but the mechanism of action is incompletely understood. It has previously been suggested that mutant *DNMT3A* causes an impaired DNA damage response in pre-leukaemic cells (Guryanova et al, 2016b). To test this, we used a model of γ-radiation induced thymic T-cell lymphoma development (Fig. 2A). In this model, repeated low dose γ-irradiation was shown to promote leukaemic transformation by acting on early haematopoietic stem and progenitor cells (HSPCs) in the bone marrow (Kaplan, 1964). These nascent leukaemic cells migrate from the bone marrow to the thymus where their expansion drives T cell lymphoma development.

We observed that *Dnmt3a^{R878H/+}* mice developed thymic T cell lymphoma significantly faster than their WT littermates (Fig. 2B).

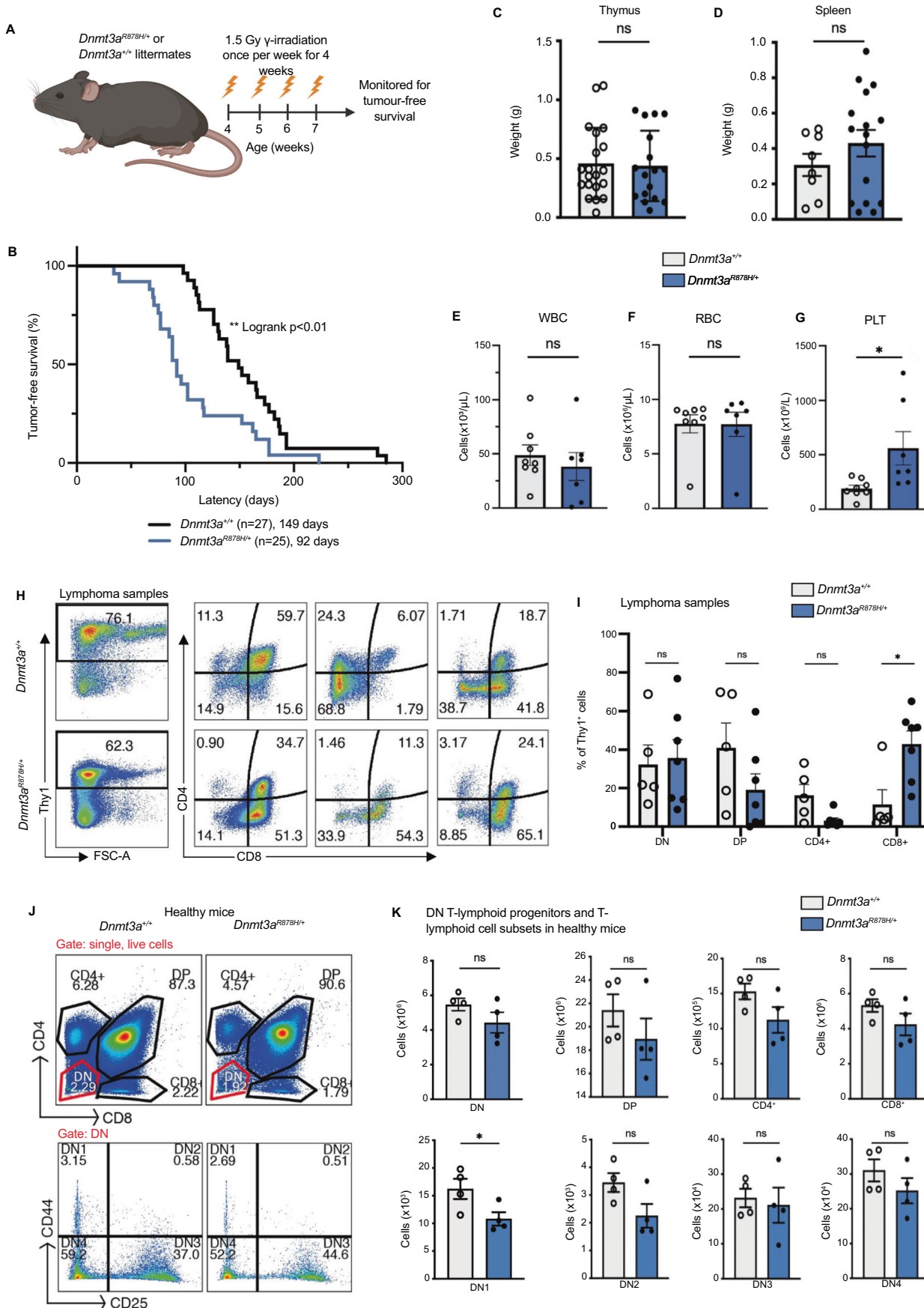

◄ **Figure 2. Haematopoietic stem and progenitor cells from $Dnmt3a^{R878H/+}$ mutant mice are primed for γ-radiation induced thymic T cell lymphoma development.**

(A) Schematic indicating $Dnmt3a^{R878H/+}$ mice and WT control littermates exposed to four doses of 1.5 Gy γ-radiation, once a week, starting the age of 4 weeks with the aim of producing γ-irradiation-induced thymic T cell lymphoma. (B) Kaplan–Meier survival curve showing tumour-free survival for $Dnmt3a^{R878H/+}$ mice and WT control littermates. (C) Thymus and (D) spleen weights respectively, of $Dnmt3a^{R878H/+}$ mutant mice and WT littermates at the ethical endpoint after developing thymic T cell lymphoma. (E) White blood cells (WBC), (F) red blood cells (RBC), and (G) platelet (PLT) counts from thymic T cell lymphoma burdened $Dnmt3a^{R878H/+}$ and WT mice at ethical endpoint. (H) Flow cytometry plots and (I) graphical representations of the distribution of CD4 and CD8 expression on thymic T lymphoma cells. Lymphomas from $Dnmt3a^{R878H/+}$ mutant mice were significantly more likely to express CD8 compared to the lymphomas from their WT littermates. This was accompanied by a modest reduction in double-positive CD4⁺CD8⁺ lymphoma cells and single positive CD4⁺ lymphoma cells although this difference did not reach statistical significance. (J) Flow cytometry gating of CD4 and CD8 single-positive T cells, double-positive (DP) T cells and double-negative (DN) T cells from healthy (i.e. non-irradiated, tumour-free) 8–10-week-old $Dnmt3a^{R878H/+}$ and control WT mice, with the DN T cells further divided into their differentiation stages (bottom; DN1 (CD25⁻CD44⁺), DN2 (CD25⁻CD44⁺), DN3 (CD25⁺CD44⁻) and DN4 (CD25⁻CD44⁻)). (K) Graphical representations of the numbers of total thymocytes, the indicated DN T lymphoid progenitors, and T lymphoid cell subsets in the thymus of $Dnmt3a^{R878H/+}$ mutant mice and their WT littermates. Data information: (B) $n = 25$ $Dnmt3a^{R878H/+}$ and $n = 27$ $Dnmt3a^{+/+}$ independent biological repeats. Statistical significance was assessed using Prism 10 software by a log rank (Mantel-Cox) statistical test; **$p < 0.01$. All error bars are the mean (±SEM) of (C, D) $n = 16$ $Dnmt3a^{R878H/+}$ and $n = 20$ $Dnmt3a^{+/+}$ independent biological repeats. Statistical significance was assessed using Prism 10 software by t tests. ns, not significant; $p > 0.05$. (E–G) $n = 8$ $Dnmt3a^{R878H/+}$ and $n = 7$ $Dnmt3a^{+/+}$ independent biological repeats. Statistical significance was assessed using Prism 10 software by t tests. ns, not significant; $p > 0.05$, *$p < 0.05$. (I) $n = 7$ $Dnmt3a^{R878H/+}$ and $n = 5$ $Dnmt3a^{+/+}$ independent biological repeats. Statistical significance was assessed using Prism 10 software by Multiple t tests. ns, not significant; $p > 0.05$, *$p < 0.05$. (K) $n = 4$ $Dnmt3a^{R878H/+}$ and $n = 4$ $Dnmt3a^{+/+}$ independent biological repeats. Statistical significance was assessed using Prism 10 software by t tests. ns, not significant; $p > 0.05$, *$p < 0.05$.

Importantly, it appeared that the $Dnmt3a^{R878H/+}$ mice had a similar tumour burden to WT mice at ethical endpoint, as indicated by similar weights of their abnormally enlarged thymus and spleen (Fig. 2C,D). Peripheral blood analysis at the point of sacrifice also revealed no significant difference in WBC and RBC counts between T cell lymphoma burdened $Dnmt3a^{R878H/+}$ and WT mice (Fig. 2E,F). However, lymphoma burdened $Dnmt3a^{R878H/+}$ mice had significantly higher platelet counts (~2-fold) compared to WT controls (Fig. 2G). No significant differences were found in other blood cell types (Fig. EV1C–H). Flow cytometric analyses of spleens revealed no significant differences in their cell composition between lymphoma burdened $Dnmt3a^{R878H/+}$ and WT controls (Fig. EV1I).

To identify possible phenotypic differences between lymphomas from $Dnmt3a^{R878H/+}$ vs WT mice, flow cytometric analysis was performed on the thymus. While there was no difference in TCRβ and B220 expression (Fig. EV1J,K), lymphomas from $Dnmt3a^{R878H/+}$ mice were significantly more likely to express CD8 compared to the lymphomas from WT mice (Fig. 2H,I). These data suggests that DNMT3A may act as a suppressor of CD8 + T cell lymphomas, and/or may regulate T cell differentiation. The latter would be in line with previous reports (Haney et al, 2016; Ladle et al, 2016). The CD8⁺ $Dnmt3a^{R878H/+}$ lymphomas may represent malignant counterparts of immature T lymphoid cells in the thymus that express CD8 prior to becoming CD4/CD8 double positive (DP).

This may indicate that mutant DNMT3A causes a defect in T cell differentiation in the thymus. To examine this, the subcellular composition of the thymus was analysed in healthy (i.e. non-irradiated, tumour-free) $Dnmt3a^{R878H/+}$ and control WT mice. There were significantly fewer double-negative differentiation stage 1 (DN1) thymocytes in $Dnmt3a^{R878H/+}$ mice compared to WT controls (Fig. 2J,K). $Dnmt3a^{R878H/+}$ thymocytes maintained a trend towards reduced cell numbers throughout the later DN stages, as well as the more mature CD4 and CD8 single positive cell subsets, however, these reductions did not reach statistical significance (Fig. 2K). Collectively, these data may indicate that mutant DNMT3A has a subtle influence on the differentiation of early T lymphoid progenitor populations in the thymus, and possibly the bone marrow.

## Serial competitive bone marrow transplantations demonstrate that $Dnmt3a^{R878H/+}$ haematopoietic stem and progenitor cells have a competitive advantage over their WT counterparts

Flow cytometric analysis of the bone marrow HSC compartments from $Dnmt3a^{R878H/+}$ mice and WT littermates (Fig. EV2A) revealed no significant differences in the numbers of granulocyte myeloid (GMP), common myeloid (CMP) or megakaryocyte erythroid (MEP) progenitors, although there was a trend towards slightly higher numbers of each of these progenitor populations in the $Dnmt3a^{R878H/+}$ mice (Fig. 3A). No significant differences were found between the numbers of long term- or short term-HSCs (LT-/ST-HSCs) between $Dnmt3a^{R878H/+}$ and WT mice (Fig. 3B). This again reveals that any effect of the $Dnmt3a^{R878H/+}$ point mutation on HSCs is rather subtle.

Previous studies have established that DNMT3A-KO HSPCs have a propensity to self-renew at the expense of differentiation (Challen et al, 2011; Guryanova et al, 2016a; Jeong et al, 2018; Mayle et al, 2015). To determine whether this holds true for HSPCs from $Dnmt3a^{R878H/+}$ mice, we conducted competitive bone marrow transplantation experiments (Fig. 3C). CD45.2 $Dnmt3a^{R878H/+}$ or WT (control) bone marrow cells were combined in a 1:1 ratio with WT CD45.1 cone marrow cells, and used to reconstitute the haematopoietic system of lethally irradiated CD45.1⁺CD45.2⁺ recipient mice. After 3 months of recovery, the proportions of progenitor cells of the different genotypes were determined based on CD45 isoform surface marker expression. While no significant difference in the proportions of $Dnmt3a^{R878H/+}$ derived cells was observed in the GMP, CMP and MEP populations, (Fig. 3D), $Dnmt3a^{R878H/+}$ LT- and ST-HSCs were strongly enriched in the bone marrow of recipient mice compared to WT derived LT- and ST-HSCs (Fig. 3E). This is in line with previous reports of $Dnmt3a^{R878H/+}$ driven clonal expansion (Loberg et al, 2019; Wang et al, 2024) and supports the notion that mutant DNMT3A impacts the balance between self-renewal and differentiation of haematopoietic stem cells.

To test whether the advantage of $Dnmt3a^{R878H/+}$ HSCs could be sustained, we isolated the bone marrow from the reconstituted recipient mice at 3 months after transplantation and repeated the

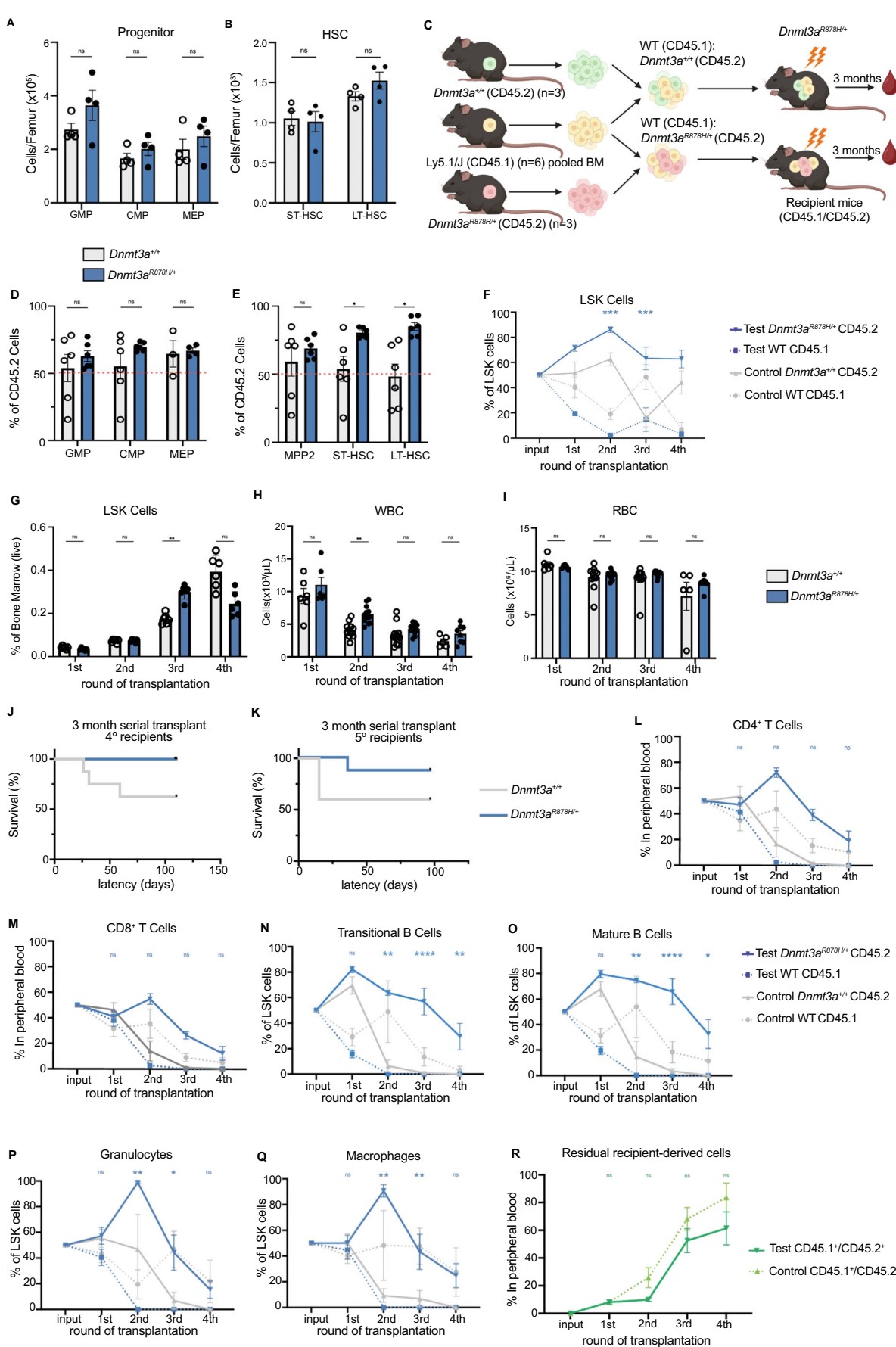

◀ **Figure 3. Haematopoietic stem and progenitor cells from *Dnmt3a^R878H/+* mutant mice have a competitive advantage over their WT counterparts.**

(A) Numbers of haematopoietic progenitor cells and (B) haematopoietic stem cells per femur from 8–12-week-old *Dnmt3a^R878H/+* mutant mice and their WT littermates. (C) Schema depicting the experimental setup for the competitive bone marrow transplantation experiments, where *Dnmt3a^R878H/+* mutant derived CD45.2 cells and *Dnmt3a^+/+* WT derived CD45.2 cells were combined in a 1:1 ratio with WT (CD45.1) cells. (D, E) The proportions of CD45.2 HSPCs (granulocyte-myeloid progenitors (GMP), common myeloid progenitors (CMP), megakaryocyte-erythroid progenitors (MEP) (D), and multipotent progenitor cells (MPP2), long-term haematopoietic stem cells (LT-HSC), and short-term haematopoietic stem cells (ST-HSC) (E) that were derived from *Dnmt3a^R878H/+* mutant donor bone marrow cells vs WT control bone marrow donor cells in recipient mice three months after bone marrow transplantation. (F) The proportions of *Dnmt3a^R878H/+* (CD45.2) donor derived LSK cells vs control WT (CD45.1) derived LSK cells in recipient mice 3 months after each round of transplantation. (G) Graph showing the proportions of donor LSK cells of *Dnmt3a^R878H/+* mutant donor origin (CD45.2) or WT donor origin (CD45.1) amongst total bone marrow cells in recipient mice that had been transplanted with a mixture of WT (CD45.1)/ *Dnmt3a^R878H/+* (CD45.2) bone marrow cells, 3 months after each round of transplantation. (H) Total numbers of WBC and (I) RBC in recipient mice 3 months after each serial transplantation with a WT:WT or a WT:*Dnmt3a^R878H/+* mixture of bone marrow cells. (J) Kaplan–Meier survival curve of fourth generation (4°) recipients of serially transplanted bone marrow of either the WT:WT or the WT:*Dnmt3a^R878H/+* mixture of bone marrow cells. Data were statistically analysed using a log-rank (Mantel-Cox) test. (K) Kaplan–Meier survival curve of fifth generation (5°) recipients of serially transplanted bone marrow of either the WT:WT or the WT:*Dnmt3a^R878H/+* mixture of bone marrow cells. Data were statistically analysed using a log-rank (Mantel-Cox) test. Blood cell proportions determined by flow cytometry for (L) CD4 + T cells, (M) CD8 + T cells, (N) transitional B Cells, (O) mature B Cells, (P) granulocytes, and (Q) macrophages, all from mice 3 months after each serial transplantation. (R) Proportions of recipient-derived cells over time in the WT:*Dnmt3a^R878H/+* reconstituted mice compared with the WT:*Dnmt3a^+/+* reconstituted control mice. Data information: All error bars are the mean (±SEM) of (A, B) $n = 4$ *Dnmt3a^R878H/+* and $n = 4$ *Dnmt3a^+/+* independent biological repeats, (D, E, G) $n = 6$ *Dnmt3a^R878H/+* and $n = 6$ *Dnmt3a^+/+* independent biological repeats, (H, I) $n = 6$ *Dnmt3a^R878H/+* and $n = 6$ *Dnmt3a^+/+* independent biological repeats (1st round), $n = 10$ *Dnmt3a^R878H/+* and $n = 10$ *Dnmt3a^+/+* independent biological repeats (2nd and 3rd rounds), $n = 8$ *Dnmt3a^R878H/+* and $n = 5$ *Dnmt3a^+/+* independent biological repeats (4th round WBC), and $n = 5$ *Dnmt3a^R878H/+* and $n = 5$ *Dnmt3a^+/+* independent biological repeats (4th round RBC). Statistical significance was assessed using Prism 10 software by t tests. ns, not significant; $p > 0.05$, $*p < 0.05$, $**p = <0.01$. (F, L–R) $n = 8$ *Dnmt3a^R878H/+* and $n = 8$ *Dnmt3a^+/+* independent biological repeats. Statistical significance was assessed using Prism 10 software by 2-way ANOVA with Šídák's multiple comparisons test. ns, not significant; $p > 0.05$, $*p < 0.0332$, $**p < 0.0021$, $***p < 0.0002$, $****p < 0.0001$.

transplantation as previously described, into lethally irradiated secondary recipient mice. This transplantation process was repeated until the recipient mice could no longer be rescued from lethal γ-irradiation. By examining the LSK (Lin⁻ Sca-1⁺ c-Kit⁺) compartment in the bone marrow after each round of transplantation we found that *Dnmt3a^R878H/+* LSK cells had a sustained competitive advantage over their WT counterparts across at least 4 rounds of serial transplantation (Fig. 3F). However, the increased proportion of *Dnmt3a^R878H/+* LSK cells did not equate to an uncontrolled proliferation of cells, as the LSK cells still made up a similar proportion of the total bone marrow when compared to mice reconstituted with WT bone marrow cells (Fig. 3G), the exception being in the third transplant, although this returned to equivalent percentages in the fourth round.

Furthermore, although LSKs from *Dnmt3a^R878H/+* mice out-competed LSK cells from WT mice in the bone marrow of reconstituted mice, the overall blood composition remained similar between the WT:WT control and the WT:*Dnmt3a^R878H/+* mixed bone marrow recipients. Mice receiving the WT:*Dnmt3a^R878H/+* bone marrow mix had significantly higher WBC counts after the second transplant, however no other significant differences were detected (Fig. 3H,I). By the 4th and 5th rounds of serial transplantation, HSC exhaustion began to cause lethal haematopoietic failure. Importantly, the mice that had received the WT:*Dnmt3a^R878H/+* bone marrow were better protected from this exhaustion compared to those transplanted with WT:WT bone marrow, as shown by the first reconstitution failure occurring only in the 5th round of transplantation (Fig. 3J,K).

When assessing the contribution of donor cells to mature lymphoid and myeloid cell subsets in the peripheral blood, *Dnmt3a^R878H/+* cells were more likely to make up a larger proportion of both CD4 and CD8 T cells compared to WT cells at all rounds of transplantation measured. However, this difference did not reach statistical significance, perhaps due to a higher number of γ-radiation resistant host-derived T cells that remained in these *Dnmt3a^R878H/+* cell recipient mice (Fig. 3L,M). Furthermore, there was a strong and sustained competitive advantage of both

transitional and mature B cells as well as granulocytes and macrophages from the *Dnmt3a^R878H/+* donors (Fig. 3N–Q). This indicates that *Dnmt3a^R878H/+* mutant HSPCs do not have a lymphoid or myeloid bias under these conditions. Across all mature haematopoietic cell types examined, it appeared as though the *Dnmt3a^R878H/+* cells lost their competitive advantage after multiple rounds of transplantation as seen by their reductions in proportion in the peripheral blood (Fig. 3L–Q). However, this coincided with an increase in the contribution of residual recipient-derived (CD45.1⁺/CD45.2⁺) lymphoid cells (Fig. 3R). These cells likely had escaped γ-radiation and reflect 'fitter' cells compared with the serially transplanted mixed bone marrow derived cells.

## *Dnmt3a^R878H/+* myeloid progenitor cells have a subtle survival advantage upon γ-irradiation

The serial competitive transplantations demonstrated that although *Dnmt3a^R878H/+* cells had a distinct survival advantage, no myeloid or lymphoid bias was seen in steady state conditions. To explore how the *Dnmt3a^R878H/+* mutation may impact the behaviour of HSCs, we analysed HSC survival after exposure to γ-irradiation. Bone marrow was harvested from 8–12-week-old *Dnmt3a^R878H/+* mice and WT littermates, cells transferred into cell culture medium (Michalak et al, 2009), and they were then either exposed to γ-irradiation (1.5 Gy) or left untreated. Cell viability was then assessed by flow cytometry over 72 h with the HSC gating strategy (Pietras et al, 2015) shown in Fig. EV2B. *Dnmt3a^R878H/+* HSCs initially trended towards slightly better survival compared to their WT HSCs, and this advantage appeared subtly more pronounced in the γ-irradiated cells (Fig. EV2C). *Dnmt3a^R878H/+* LSK cells showed a trend towards better survival compared to their WT counterparts at 12 h, although no survival advantage was observed at later timepoints (Fig. EV2D). No statistically significant differences were found between the HSC compartments when when looking at total cell numbers across all timepoints. Non-irradiated *Dnmt3a^R878H/+* ST- and LT-HSCs trended toward better survival than their WT counterparts, though this did not reach statistical significance

(Fig. EV3A,B). The $Dnmt3a^{R878H/+}$ myeloid progenitor cells (MPP3) were observed to be slightly more robust following γ-irradiation compared to their WT counterparts although this was also not statistically significant (Fig. EV3D). These data support the notion that $Dnmt3a^{R878H/+}$ HSPCs have a subtle competitive advantage, with a similarly subtle survival advantage seen in myeloid progenitors upon DNA damage.

## Transcriptomic profiling of γ-irradiated pre-leukaemic $Dnmt3a^{R878H/+}$ LSK cells reveals subtle downregulation of $p53$ signalling pathway genes

We have demonstrated that $Dnmt3a^{R878H/+}$ mice succumb more rapidly to γ-irradiation induced thymic lymphoma. We hypothesised this may be due to an abnormal or incomplete stress response and therefore explored the transcriptional changes occurring in response to γ-irradiation in $Dnmt3a^{R878H/+}$ and WT LSK cells by conducting RNA-seq analysis. $Dnmt3a^{R878H/+}$ and WT littermates were either left untreated or exposed to low dose, whole body γ-irradiation (2 Gy), and then allowed to recover for 2 h before bone marrow was harvested and LSK cells were isolated by FACS. The RNA was then extracted from LSK cells and bulk RNA-seq was performed (Fig. 4A). When comparing transcriptional changes between LSK cells from γ-irradiated $Dnmt3a^{R878H/+}$ LSK cells with their γ-irradiated WT counterparts, 335 genes were observed to be differentially expressed (DE), with 105 downregulated and 230 upregulated DE genes identified (Fig. 4B,C).

Many genes were observed to be differentially expressed between $Dnmt3a^{R878H/+}$ and WT LSK cells at baseline (Fig. EV4B). As expected, the $p53$ signalling pathway was significantly enriched in both $Dnmt3a^{R878H/+}$ and WT LSK cells upon γ-irradiation. However, upon γ-irradiation, when compared to WT LSK cells the $p53$ signalling pathway was induced to a significantly lesser extent in $Dnmt3a^{R878H/+}$ LSK cells, suggesting a blunted $p53$ response to DNA damage (Fig. 4D). GO pathway analysis further revealed a significant upregulation of gene transcriptional networks related to oxidative phosphorylation and various mitochondrial complex formation pathways in γ-irradiated $Dnmt3a^{R878H/+}$ LSK cells compared to their WT counterparts (Fig. 4E). However, Seahorse mito stress assays revealed no significant difference between the maximal oxygen consumption rate (OCR) of $Dnmt3a^{R878H/+}$ myeloid or B cells compared to their WT counterparts (Fig. EV5A–D).

## $Dnmt3a^{R878H/+}$ does not accelerate tumour onset of $Trp53^{+/-}$ heterozygous mice

$Dnmt3a^{+/+}/Trp53^{+/-}$ and $Dnmt3a^{R878H/+}/Trp53^{+/-}$ mice were generated, to observe the combined impact of loss of one $Trp53$ allele and the $Dnmt3a$ mutation on tumorigenesis. No significant difference was observed in overall and tumour-free survival when the $Dnmt3a$ mutation was introduced (Fig. 5A). This indicates that the $Dnmt3a^{R878H/+}$ mutation does not cooperate with the loss of one allele of $Trp53$ to accelerate tumour development. Existing sequencing datasets of human AML samples demonstrate that mutations in $DNMT3A$ and $TP53$ do not usually co-occur (Weinstein et al, 2013), and this mutual exclusivity suggests that mutations in these genes may have overlapping functions in the development of malignant disease.

The RNA sequencing data shows that $p53$ gene expression is subtly downregulated in $Dnmt3a^{R878H/+}$ LSK cells following γ-irradiation (Fig. 5B). Genes within the $p53$ pathway were identified, and a t-test was performed for each gene, to compare transcript enrichment between γ-irradiated $Dnmt3a^{R878H/+}$ vs γ-irradiated WT LSKs. While none of the genes were found to have a statistically significant difference between the two genotypes, many genes were subtly downregulated in the γ-irradiated $Dnmt3a^{R878H/+}$ LSK cells compared to their WT counterparts (Fig. EV6A). The consistency of slight downregulation of multiple $p53$ pathway genes suggests that while subtle, this effect is not random. Of note, $p53$ pathway dysregulation may result in broader transcriptional disruption of downstream (e.g. indirect target) genes (Rocha et al, 2003).

Indeed, the RNA sequencing data reveal dysregulation of genes that are not direct $p53$ targets. For instance, CyclinD1 ($Ccnd1$) and CyclinD2 ($Ccnd2$) are key regulators of cell cycle progression and survival, and the overexpression of these genes has previously been implicated in certain haematologic malignancies (Metcalf et al, 2010). Although not statistically significant, we detected clear enrichment of $Ccnd1$ and $Ccnd2$ mRNA in γ-irradiated $Dnmt3a^{R878H/+}$ LSK cells (Fig. 5C,D). This finding, alongside an increase in mRNA for the pro-survival Bcl-2 protein $Mcl-1$ (Fig. 5E), and reduced expression of DNA damage response genes ($Chek1$, $Chek2$, $Rad51$) (Fig. 5F–H), all points to a complex and disrupted cell death regulatory environment in γ-irradiated $Dnmt3a^{R878H/+}$ LSK cells.

$Puma/Bbc3$ is well established as a transcriptional target of $p53$. When the $p53$ signalling pathway is initiated, $Puma/Bbc3$ is transcriptionally activated resulting in an increase in PUMA protein expression (Nakano and Vousden, 2001). Notably, our RNA sequencing data show that there was no difference in $Puma/Bbc3$ mRNA levels between $Dnmt3a^{R878H/+}$ and WT LSK cells, both at baseline or after γ-irradiation (Figs. 4B and EV4B). To further explore the weaker p53 pathway signalling observed in transcriptomic analysis of γ-irradiated $Dnmt3a^{R878H/+}$ LSK cells, and to functionally measure $p53$ dysregulation with a more sensitive assay which can measure beyond just the transcript level, we utilized a $Puma-tdTomato$ reporter mouse (Lieschke et al, 2024) in which the coding sequences for tdTomato have been knocked into the $Puma/Bbc3$ gene locus. $Puma/Bbc3$ is a direct transcriptional target of $p53$ which is critical for $p53$ induced apoptosis (Villunger et al, 2003). Evidence indicates that $p53$ is the primary driver of $Puma/Bbc3$ induction in response to DNA damage, particularly following γ-irradiation (Jeffers et al, 2003; Kuchur et al, 2021). While $Puma/Bbc3$ can be regulated by other transcription factors in response to different apoptotic stimuli, such as treatment with glucocorticoids, in this scenario of γ-irradiation induced DNA damage, $p53$ is the predominant driver of induction of $Puma/Bbc3$ transcription (Yu and Zhang, 2008). Thus, $Puma$ reporter expression was used as a measure of $p53$ transcriptional activity. Of note, the regulatory elements (5' and 3' UTRs) of the $Puma/Bbc3$ gene are still present in the $Puma-tdTomato$ reporter mice, and so the mRNA undergoes similar transcriptional and post-transcriptional regulation as endogenous $Puma/Bbc3$ mRNA. Since tdTomato replaces the $Puma/Bbc3$ coding sequence, the post-translational modifications do not occur on the tdTomato protein. Therefore, tdTomato expression in this model serves as a transcriptional readout of $Puma$ induction but does not reflect post-translational regulation of the $Puma/Bbc3$ protein itself. Male $Dnmt3a^{R878H/+}$ mice were crossed with female $Puma-tdTomato^{KI/KI}$ reporter mice, and at 8–12 weeks, $Dnmt3a^{R878H/+}/Puma-tdTomato^{KI/+}$ and $Dnmt3a^{+/+}/Puma-tdTomato^{KI/+}$ littermates were

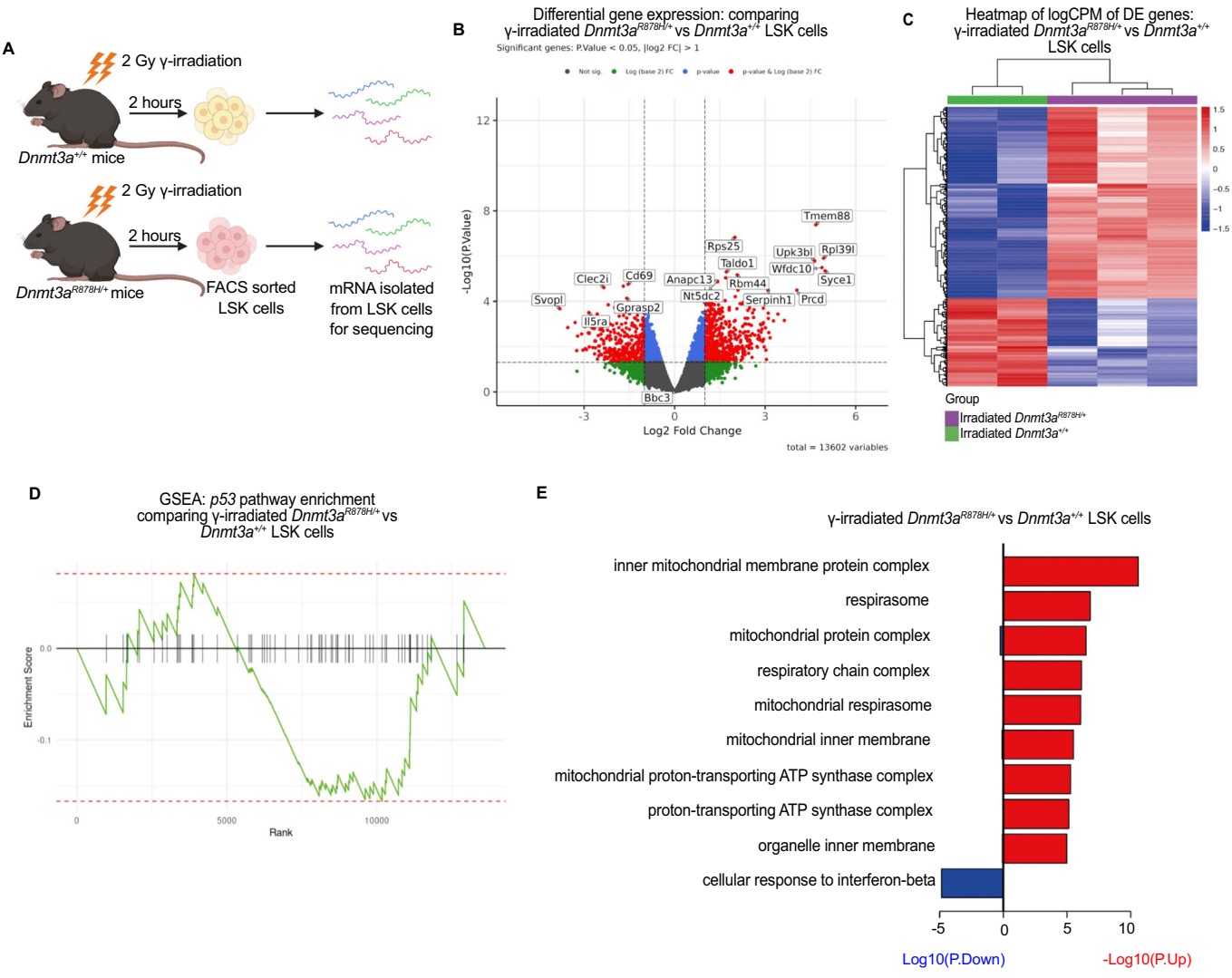

**Figure 4. RNA sequencing of γ-irradiated *Dnmt3a^R878H/+* LSK cells reveals transcriptomic alterations including p53 pathway dysregulation.**

(A) Schematic depicting the experimental protocol where *Dnmt3a^R878H/+* and WT control littermates were exposed to a single dose of whole-body γ-radiation (2 Gy). The mice were allowed to recover for 2 h before LSK cells were isolated from the bone marrow for RNA-seq analysis. (B) Volcano plot depicting gene expression differences between LSK cells harvested from γ-irradiated *Dnmt3a^R878H/+* mice *vs* γ-irradiated WT control mice, where genes in red are significantly differentially expressed, and ranked by *p*-value and log fold change. Genes meeting the significance threshold are labelled. *Puma/Bbc3* has been highlighted. (C) Hierarchical clustering heatmap depicting differentially expressed genes between γ-irradiated *Dnmt3a^R878H/+* LSK cells *vs* γ-irradiated WT control LSK cells. Red and blue squares indicate genes with high or low gene expression levels, respectively. (D) Gene Set Enrichment Analysis (GSEA) plot comparing *p53* pathway gene expression between γ-irradiated *Dnmt3a^R878H/+* LSK cells vs γ-irradiated WT LSK cells, with the majority of genes in this pathway being downregulated (negative enrichment score) in the *Dnmt3a^R878H/+* LSK cells. (E) The top 10 significantly enriched GO terms associated with differentially expressed genes in γ-irradiated *Dnmt3a^R878H/+* LSK cells compared to γ-irradiated WT control LSK cells. Data information: (B, E) $n = 3$ *Dnmt3a^R878H/+* and $n = 2$ *Dnmt3a^+/+* independent biological repeats. Statistical significance criteria is $p < 0.05$, $\log_2 FC > 1$. Statistical significance was determined using the limma-trend method with empirical Bayes moderation after fitting a linear model to log-transformed counts per million (logCPM). Surrogate variable analysis (SVA) was incorporated to adjust for hidden sources of variation. GO enrichment analysis was conducted using goana.

either exposed to γ-irradiation (5 Gy) or left untreated. 8 h post-treatment, the spleen, thymus and bone marrow cells of the mice were collected, and tdTomato expression was measured by flow cytometry to observe *Puma* induction as a readout of *p53* activity (Fig. 6A). As expected, γ-irradiation led to *Puma* reporter induction in all tissues examined in both *Dnmt3a^R878H/+* and WT mice (Fig. 6B–D). The WT spleen and bone marrow cells showed significantly higher *Puma* reporter induction following γ-irradiation compared to their *Dnmt3a^R878H/+* counterparts (Fig. 6E,F), while no significant difference was observed in the thymus (Fig. EV7). Although we found no

significant difference in *Puma/Bbc3* mRNA levels between LSK cells from *Dnmt3a^R878H/+* and WT mice by sequencing, the more sensitive reporter model was able to demonstrate a subtle functional impairment in *Puma/Bbc3* induction upon *p53* activation in certain tissues from *Dnmt3a^R878H/+* mice. This observed reduction in *Puma/Bbc3* transcriptional reporter activity in *Dnmt3a^R878H/+* cells strongly aligns with the notion of attenuated *p53* activity following γ-irradiation.

Taken together, these data support the notion that *p53* is dysregulated in *Dnmt3a^R878H/+* cells. This disrupted *p53* activity may

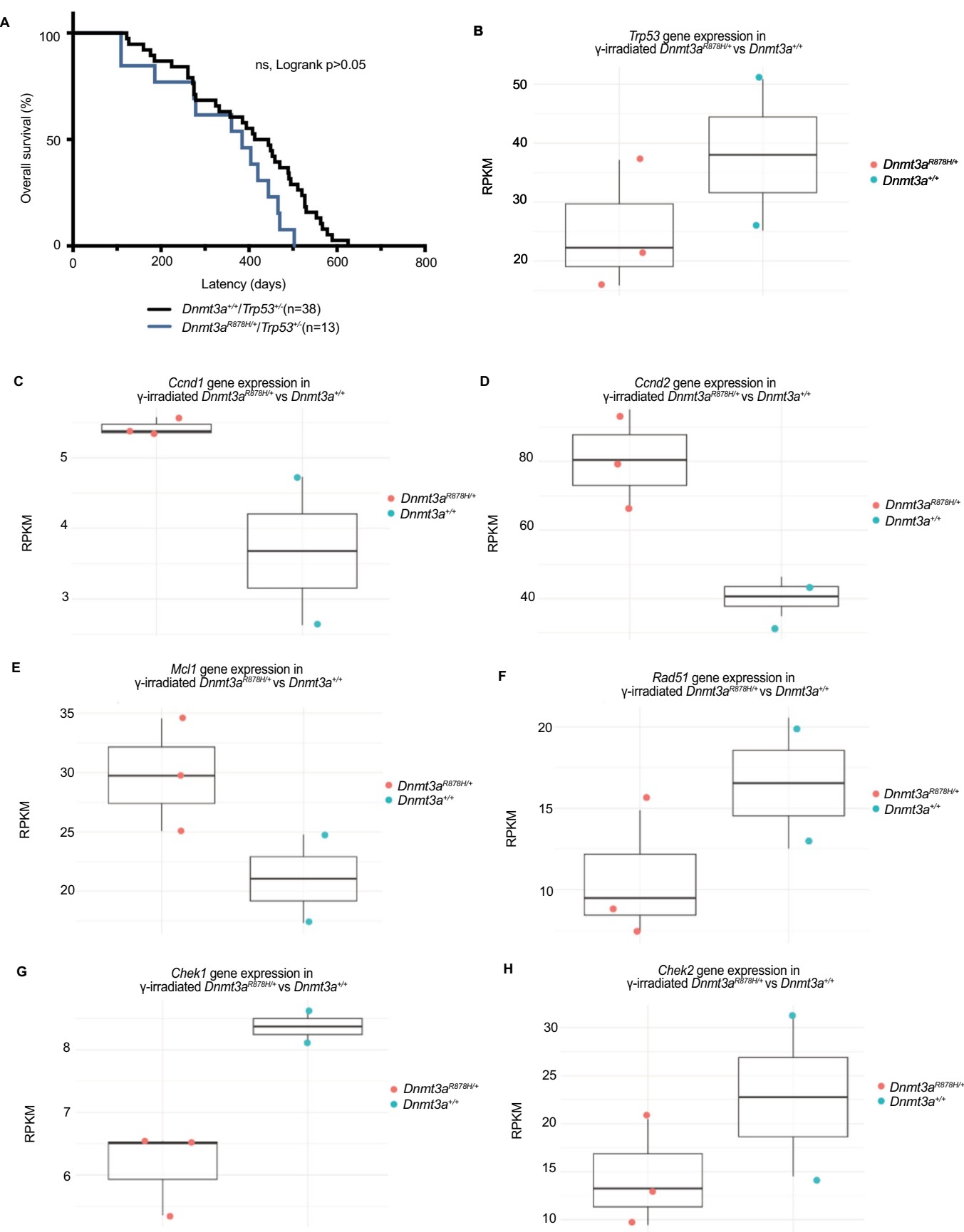

◀ **Figure 5.  Lack of accelerated tumour development in *Dnmt3a*$^{R878H/+}$/*Trp53*$^{+/-}$ mice and dysregulated transcription of *p53* pathway genes suggests an impaired *p53* response in *Dnmt3a*$^{R878H/+}$ cells.**

(A) Kaplan–Meier survival curve displaying tumour-free survival of *Dnmt3a*$^{+/+}$/*Trp53*$^{+/-}$ mice and *Dnmt3a*$^{R878H/+}$/*Trp53*$^{+/-}$ mice. Mice were excluded if cause of death was not attributed to a tumour e.g. euthanasia ethically mandated due to malocclusion, or wounds from fighting. (B–H) Box plots showing normalised expression of select key *p53* pathway genes, and the pro-survival *Bcl-2* protein Mcl-1, in γ-irradiated (2 Gy) *Dnmt3a*$^{R878H/+}$ LSK cells vs γ-irradiated (2 Gy) Dnmt3a$^{+/+}$ LSK cells, in reads per kilobase per million mapped reads (RPKM), (B) *Trp53*, (C) *Ccnd1*, (D) *Ccnd2*, (E) *Mcl-1*, (F) *Rad51*, (G) *Chek1*, (H) *Chek2*. Data information: (A) n = 13 *Dnmt3a*$^{R878H/+}$/*Trp53*$^{+/-}$ and n = 38 *Dnmt3a*$^{+/+}$/*Trp53*$^{+/-}$ independent biological repeats. Statistical significance was assessed using Prism 10 software by a log rank (Mantel-Cox) statistical test; ns, not significant; p > 0.05. (B–H) n = 3 *Dnmt3a*$^{R878H/+}$ and n = 2 *Dnmt3a*$^{+/+}$ independent biological repeats. Each box represents the interquartile range (IQR), with the lower and upper bounds corresponding to the 25th and 75th percentiles, respectively. The centre line indicates the median (50th percentile). Whiskers extend to the minimum and maximum values within 1.5× IQR; data points beyond this range are considered outliers.

afford mutant cells a subtle advantage when sustaining DNA damage, and this may prime these cells for clonal expansion.

## Discussion

There is a growing body of evidence for the importance of *DNMT3A* in haematopoiesis, showing that impairment of its function can initiate CH and leukaemogenesis (Lu et al, 2016; Mayle et al, 2015; Yang et al, 2015b; Zhang et al, 2020). In this study, we generated a mouse model that replicates the most common human *DNMT3A* mutation, the R882H substitution, by introducing the homologous murine mutation, R878H, into the *Dnmt3a* locus using CRISPR/Cas9 gene editing technology. An advantage of this model is that *Dnmt3a*$^{R878H/+}$ cells can be studied without interference from Cre mediated gene recombination. This strategy allowed us to study the impact of *Dnmt3a*$^{R878H/+}$ on haematopoiesis in intact mice and allowed us to isolate and study cell populations without the need for Cre mediated induction of the mutation. The data presented here support and extend findings from previous studies in which expression of *Dnmt3a*$^{R878H}$ was studied in the context of normal and malignant haematopoiesis (Dai et al, 2017). *DNMT3A* R882 mutations are now widely reported to be among the most common driver mutations in patients with AML (Jawad et al, 2022; Ley et al, 2010; Yang et al, 2015a). However, as genomic sequencing becomes a mainstay in cancer diagnosis and research, mutant *DNMT3A* has also been found to occur in a range of lymphoid malignancies, particularly among patients with adult T-ALL (Bond et al, 2019; Kramer et al, 2017).

A fast and efficient low coverage DNA sequencing approach revealed that our *Dnmt3a*$^{R878H/+}$ mice have reduced methylation across all tissues examined. This hypomethylation phenotype is particularly pronounced in the brain, and less pronounced in lymphoid tissues. It has been shown that loss or alteration of *Dnmt3a* combined with a secondary cooperating oncogenic lesion can induce leukaemia in mice (Chang et al, 2015; Meyer et al, 2016; Yang et al, 2016), but precisely how mutant DNMT3A contributes to the initiation of leukaemia is still only poorly defined. There is also not yet a consensus in the field as to how the *DNMT3A* R882 hotspot mutation differs from deletion of the *DNMT3A* gene. Some groups proposed that mutant *DNMTA* R882 acts in a dominant negative manner (Kim et al, 2013; Russler-Germain et al, 2014), whereas others suggested that the hotspot mutation may alter the flanking sequence preference of *DNMT3A* (Emperle et al, 2019; Emperle et al, 2018). Realistically, it could be a contribution of both, alongside other features of *DNMT3A* R882 that have yet to be

defined. It has been suggested that *DNMT3A* mutations can interfere with the ability of cells to sense and repair DNA damage, leading to increased mutagenesis, thereby driving leukaemogenesis (Guryanova et al, 2016b).

We were able to test this hypothesis using our CRISPR engineered *Dnmt3a*$^{R878H/+}$ mice, as we could directly compare the effects of γ-irradiation induced DNA damage in vivo. In line with previous studies (Buscarlet et al, 2017), we found that the *Dnmt3a*$^{R878H}$ mutation had only minimal impact on haematopoiesis at steady state. However, we observed a significant acceleration of γ-radiation induced thymic lymphoma development in *Dnmt3a*$^{R878H/+}$ mice compared to WT littermates. We know from previous work that the cell of origin of thymic lymphoma is a bone marrow derived progenitor cell (Kaplan, 1964). This indicates that the accelerated T cell lymphoma development in the *Dnmt3a*$^{R878H/+}$ animals likely is due to interference with the stress response in bone marrow HSPCs. Indeed, the work of others has shown that *Dnmt3a*$^{-/-}$ LSK cells exhibit a pre-leukaemic phenotype (Mayle et al, 2015). Moreover, previous work on other murine models of *Dnmt3a*$^{R878H/+}$ LSK cells have also demonstrated pre-leukaemic traits in these cells (Dai et al, 2017). Collectively, these studies suggest that mutant *DNMT3A* could drive a proliferative phenotype in HSCs at the expense of differentiation. Therefore, we assessed whether we could also observe such a phenotype in our CRISPR generated *Dnmt3a*$^{R878H/+}$ mutant mice.

Interestingly, we did not find any accumulation of *Dnmt3a*$^{R878H/+}$ HSPCs in the bone marrow of 8-week-old mice, indicating that any proliferative advantage conferred by mutant *DNMT3A* is subtle. To examine the fitness of *Dnmt3a*$^{R878H/+}$ HSCs we performed serial competitive bone marrow transplant experiments. This revealed that *Dnmt3a*$^{R878H/+}$ HSCs accumulate in the bone marrow, while any changes in the proportions of mature haematopoietic cell subsets in peripheral blood are only detectable 3 months after these transplantations. This finding supports previous research which revealed that *Dnmt3a*$^{R878H/+}$ LSK cells can give rise to an increased abundance of *Dnmt3a*$^{R878H/+}$ mature leucocytes in peripheral blood (Xu et al, 2014). Similarly, in competitive bone marrow reconstitution experiments, we observed an increase in the proportions of *Dnmt3a*$^{R878H/+}$ HSC derived lymphoid and myeloid cells in peripheral blood compared to WT HSC derived cells. This showed that *Dnmt3a*$^{R878H/+}$ HSCs and their descendants possess a survival advantage over their WT counterparts in the myeloid lineage. Our results further suggest that upon DNA damage, *Dnmt3a*$^{R878H/+}$ MPP3s cells have slightly increased survival than their WT counterparts. Although the difference was not statistically significant, this finding may indicate that *Dnmt3a*$^{R878H/+}$ MPP3s cells are better able than their WT counterparts to survive and

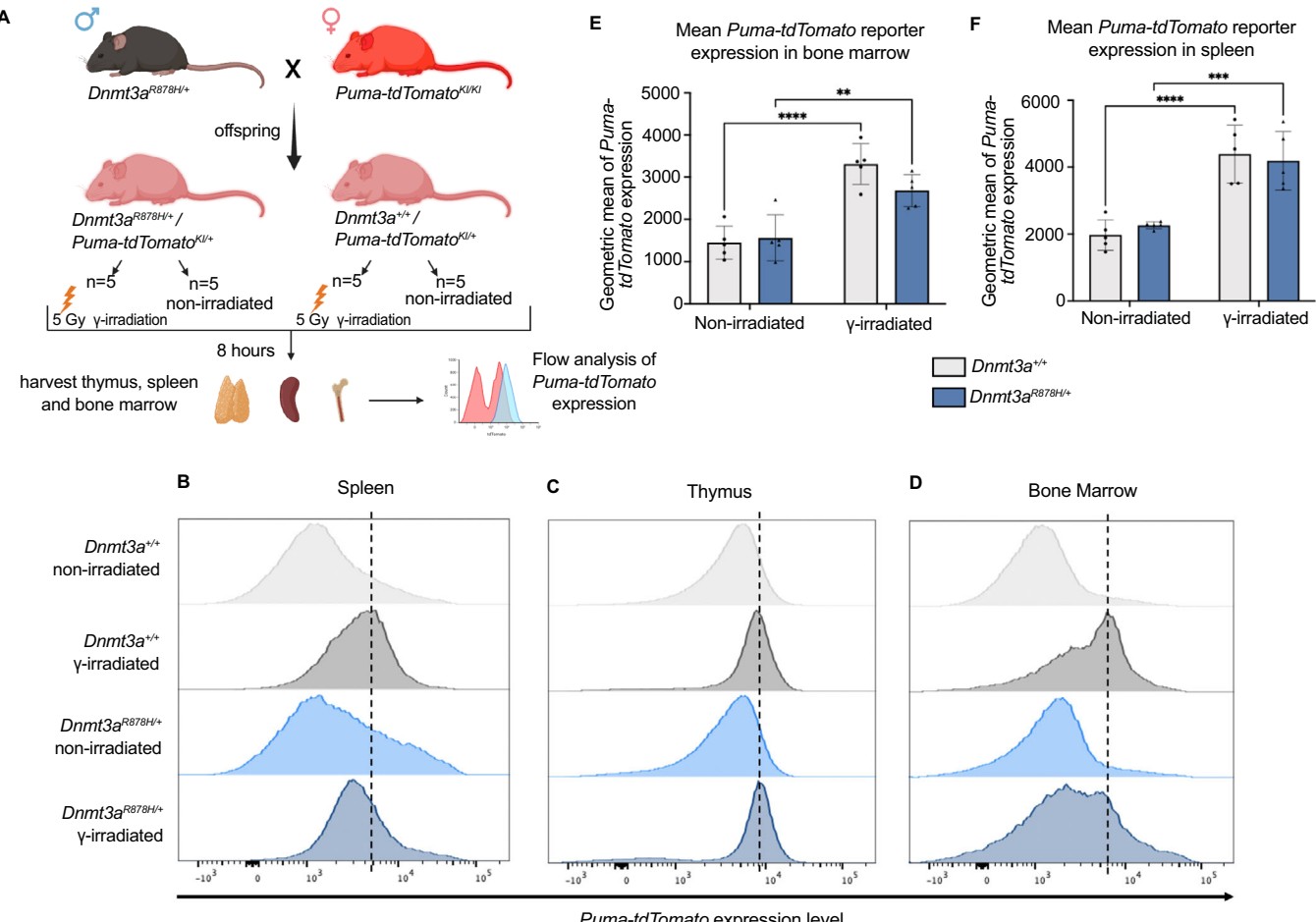

**Figure 6. There is reduced *Puma-tdTomato* reporter induction in γ-irradiated *Dnmt3a^R878H/+* spleen and bone marrow cells.**

(A) Schema showing the mating of *Dnmt3a^R878H/+* males with *Puma-tdTomato* females and experimental layout. *Dnmt3a^R878H/+*/*Puma-tdTomato^KI/+* and *Dnmt3a^+/+*/*Puma-tdTomato^KI/+* offspring were either treated with 5 Gy γ-irradiation or were left untreated. The thymus, spleen, and bone marrow from each mouse were then harvested after 8 h and analysed for tdTomato expression by flow cytometry. (B–D) Representative histograms showing tdTomato expression (reporting on *Puma/Bbc3* mRNA) in spleen (B), thymus (C) and bone marrow (D) cells in the γ-irradiated and non-irradiated mice of the indicated genotype. (E, F), Graphical representations of geometric means of tdTomato expression in the bone marrow (E) and spleen cells (F) from mice of the indicated treatment and genotype. Data information: *n* = 5 γ-irradiated *Dnmt3a^R878H/+*/*Puma-tdTomato^KI/+*, *n* = 5 non-irradiated *Dnmt3a^R878H/+*/*Puma-tdTomato^KI/+*, *n* = 5 γ-irradiated *Dnmt3a^+/+*/*Puma-tdTomato^KI/+*, and *n* = 5 non-irradiated *Dnmt3a^+/+*/*Puma-tdTomato^KI* independent biological repeats. (E, F) Error bars are the mean (±SEM). Statistical significance was assessed using Prism 10 software by 2-way ANOVA with Šídák's multiple comparisons test, **$p < 0.01$, ***$p < 0.001$, ****$p < 0.0001$.

proliferate to compensate for the cell death seen in other stem cell populations as well as the mature haematopoietic cell compartments. As MPP3 cells proliferate, the acquisition of additional mutations over time could contribute to the development of haematological cancers.

To explore how this proliferative advantage of *Dnmt3a^R878H/+* HSPCs could cause the increased propensity for leukaemia development in the γ-radiation induced thymic lymphoma model, we performed a transcriptomic analysis. RNAseq revealed weakened activation of the *p53* pathway in *Dnmt3a^R878H/+* LSK cells compared to their WT counterparts following γ-irradiation. Key genes in the *p53* pathway, including *Trp53*, *Ccnd1*, and *Ccnd2* were found to have aberrant transcript expression in *Dnmt3a^R878H/+* LSKs following γ-irradiation. The generation of *Dnmt3a^R878H/+*/*Trp53^+/−* mice showed that mutant *Dnmt3a* does not cooperate with loss of one allele of *p53* to drive tumour development. This indicates that

the *Dnmt3a^R878H/+* mutation causes *p53* pathway dysregulation such that removing one allele of *p53* might not produce additive effects to drive tumorigenesis, because the pathway is already impaired. A *Puma-tdTomato* reporter mouse model was used to verify this impaired p53 transcriptional response. Reporter expression was significantly blunted following γ-irradiation in *Dnmt3a^R878H/+* bone marrow and spleen cells compared to their counterparts from WT mice.

Collectively, our work confirms the *Dnmt3a^R878H/+* mutation causes a broad hypomethylation phenotype and drives the expansion of pre-leukaemic HSPCs over time, as seen in our successive bone marrow transplantation studies. Our findings further show that the *Dnmt3a^R878H/+* mutation drives altered gene expression patterns in LSK cells that have sustained DNA damage, providing them with a proliferative and survival advantage over their WT counterparts. We propose that this advantage is driven at

least in part by the attenuation of *p53* pathway activity after exposure to stress stimuli. The weakened *p53* response may contribute to the development of CH and leukaemia. Therefore, boosting the *p53* pathway should be considered as an approach for the treatment of CH and leukaemias driven by mutant *DNMT3A*.

# Methods

## Reagents and tools table

| Mouse model | Source | | |
|---|---|---|---|
| *Dnmt3a*$^{R878H/+}$ (*M. musculus*) | MAGEC lab (WEHI), This study | | |
| *Puma-tdTomato* (*M. musculus*) | MAGEC lab (WEHI), (Lieschke et al, 2024) | | |
| C57BL/6 (*M. musculus*) | WEHI | | |

**Antibodies and viability markers**

| Antibody-Fluorochrome | Clone | Source | Catalogue number |
|---|---|---|---|
| CD4-PerCPCy5.5 | GK1.5-7 | BioLegend | 317427 |
| CD4-FITC | H129.19 | BioLegend | 130308 |
| CD4-A700 | GK1.5 | WEHI | N/A |
| CD8-BV650 | 53-6.7 | BD Biosciences | 563234 |
| CD8-A700 | 53-6.7 | WEHI | N/A |
| TCRβ-PE-Cy7 | H57.597 | BioLegend | 109222 |
| B220-BV605 | RA3-6B2 | BioLegend | 103244 |
| B220-A700 | RA3-6B2 | BioLegend | 103231 |
| MAC1/CD11b-APC-Cy7 | M1/70 | BD Biosciences | 557657 |
| GR1-FITC | RB6-8C5 | BioLegend | 108405 |
| GR1-A700 | RB6-8C5 | WEHI | N/A |
| cKIT/CD117-BV711 | 2B8 | BD Biosciences | 563160 |
| cKIT/CD117-APC | 2B8 | WEHI | N/A |
| SCA1-A594 | E13-161.7 | BioLegend | 122501 |
| SCA1-PE-Cy7 | D7 | BD Biosciences | 558162 |
| CD19-FITC | ID3 | WEHI | N/A |
| TER119-PE | TER119 | BioLegend | 116207 |
| TER119-A700 | TER119 | WEHI | N/A |
| IgD-BV510 | 11.26c.2a | BD Biosciences | 563110 |
| IgD-FITC | 11.26c.2a | BioLegend | 405703 |
| NK1.1-A700 | PK136 | BioLegend | 108729 |
| CD44-FITC | IM781 | BioLegend | 103021 |
| CD25-PE | PC61 | BioLegend | 102007 |
| Ly6G-A700 | 1A8 | WEHI | N/A |
| CD2-A700 | RM2-5 | WEHI | N/A |
| CD3-PE | 14-2C11 | BioLegend | 100307 |
| CD3-A700 | KT3-1-1 | WEHI | N/A |
| CD16/32-PerCPCy5.5 | 93 | BioLegend | 101323 |

| Mouse model | Source | | |
|---|---|---|---|
| IgM-APC | MM-30 | BioLegend | 401613 |
| IgM-PE-Cy5 | RMM-1 | BioLegend | 406543 |
| CD34-FITC | RAM34 | BD Biosciences | 553733 |
| CD135-PE | A2F10 | BioLegend | 135305 |
| CD48-FITC | HM48-1 | BD Biosciences | 557484 |
| CD150-BV421 | TC15-12F12.2 | BioLegend | 115925 |
| CD45.1-BV421 | A20 | BioLegend | 110731 |
| CD45.2-PE | 104 | BioLegend | 109807 |
| F4/80-A700 | BM8 | BioLegend | 123129 |
| PI | N/A | Sigma-Aldrich | P4864 |
| DAPI | N/A | Merck | D9542 |
| Fluorogold | N/A | Bio-Strategy | SANTSC-35883 |
| ViaDye Red | N/A | Cytek Biosciences | R7-60008 |

**Consumables and reagents**

| Reagent/resource | Source | Catalogue number |
|---|---|---|
| Zymo Quick-DNA™ Miniprep Plus Kit | Zymo | D4069 |
| Oxford Nanopore Technologies Rapid Barcoding kit | Oxford Nanopore Technologies | SQK-RBK114-96 |
| R10.4.1 PromethION flow cell | Oxford Nanopore Technologies | FLO-PRO114M |
| MagniSort Streptavidin Negative Selection Beads | Thermo Fisher | MSNB-6002-74 |
| miRNeasy kit | Qiagen | 217084 |
| DNase digestion | Qiagen | 79254 |
| Nextera XT DNA Library prep kit | Illumina | 20027213 |
| DPBS | Gibco | |
| FBS | Gibco | |
| RPMI | Thermo Fisher | |

**Equipment**

| Machine | Source |
|---|---|
| ADVIA 2120 | Siemens |
| ADVIA 2120i | Siemens |
| Nanopore sequencer | Oxford Nanopore Technologies |
| LSRII | BD Biosciences |
| Fortessa1 | BD Biosciences |
| Fortessa X20 | BD Biosciences |
| Aurora | Cytek |
| FACSAria III | BD Biosciences |
| FACSAria Fusion 9 | BD Biosciences |
| Nanodrop | Thermo Fisher Scientific |
| XFe96 Extracellular Flux Analyzer | Agilent Technologies |

| Mouse model | Source |
| --- | --- |
| **Software** | |
| FlowJo v10 | BD Biosciences |
| Graphpad Prism 10 | GraphPad |
| base R (v4.4.1) | rOpenSci |

## Animals

All experiments with mice were approved by the Walter and Eliza Hall Institute of Medical Research (WEHI) animal ethics committee. The *Dnmt3a*$^{R878H/+}$ mice and *Puma-tdTomato* reporter mice (Lieschke et al, 2024) were generated by the MAGEC laboratory at The Walter and Eliza Hall Institute (WEHI) as previously described by Kueh et al (Kueh et al, 2017). These mice were generated and maintained on a C57BL/6 background. For the *Dnmt3a*$^{R878H/+}$ mice, Cas9 mRNA, a *Dnmt3a*-targeting single guide RNA (sgRNA) (5′-CCTCGCCAAGCGGCT CATGT-3′), and donor reference oligo (5′-ccgcactcactcccttccctg ccttcctcccacagGGTGTTTGGCTTCCCCGTCCACTACACCGACGTC TCTAACATGAGCCACTTGGCGAGGCAGAGACTGCTGGGCCG ATCGTGGAGCGTGCCGGTCATCCGCCACCTCT-3′) were injected into fertilised single-cell stage embryos. Two-cell stage embryos were then transplanted into pseudo-pregnant female C57BL/6 mice. Viable offspring were genotyped by next-generation sequencing. The *Dnmt3a*$^{R878H/+}$ mice were continually backcrossed with C57BL/6 mice to prevent the occurrence of epigenetic drift. The generation of the *Trp53*$^{+/-}$ mice has been previously described (Jacks et al, 1994). These mice had been backcrossed with C57BL/6 mice for >20 generations at The Walter and Eliza Hall Institute prior to the commencement of the studies described her. Female *Trp53*$^{+/-}$ mice were used to generate compound heterozygous *Dnmt3a*$^{R878H/+}$*Trp53*$^{+/-}$ mice by inter-crossing them with male *Dnmt3a*$^{R878H/+}$ mice; this was done because of breeding difficulties of female *Dnmt3a*$^{R878H/+}$ mutant mice. Generation of the Puma-tdTomato reporter mice, produced and maintained on a C57BL/6 background, has been previously described (Lieschke et al, 2024).

## γ-radiation induced T cell lymphoma

Starting at the age of 4 weeks, *Dnmt3a*$^{R878H/+}$ mice or WT control littermates were administered low dose (1.5 Gy) γ-irradiation, once a week for 4 weeks (total 4 doses). Trained technicians, blind to mouse genotype, identified and euthanised mice that had reached the predetermined ethical endpoint. Subsequently, thymic T cell lymphoma was diagnosed by abnormally increased thymus weight and/or abnormal CD4 and CD8 expression on abnormally enlarged cells in the thymus and/or spleen. Lymphoma-free survival was calculated in days from the last dose of γ-irradiation.

## Peripheral blood analysis

Peripheral blood was collected by retro-orbital or mandible venous puncture from live mice, or by heart puncture of mice at time of sacrifice. Peripheral blood was dispensed into EDTA-coated microvettes (Sarstedt, 20.1288), and 200 μL of peripheral blood was then analysed using an ADVIA 2120 or 2120i apparatus (Siemens).

## Low coverage DNA sequencing for methylation analyses

Body tissues and select brain regions were harvested from 4- to 5-month-old *Dnmt3a*$^{R878H/+}$ mice or WT control littermates. DNA extraction was performed using a Zymo Quick-DNA™ Miniprep Plus Kit (Zymo, D4069). DNA quantification was performed using a NanoDrop, before samples were equalised to 10 ng/μL. Libraries were prepared according to the manufacturer's instructions using the Oxford Nanopore Technologies Rapid Barcoding kit (SQK-RBK114-96). The libraries were then loaded onto a R10.4.1 PromethION flow cell (FLO-PRO114M) and sequenced at 0.01–0.05X coverage on a Nanopore sequencer, following the Skim-seq method (Faulk, 2023). Base calling was performed with Dorado 0.7.0 (https://github.com/nanoporetech/dorado) using the super-accuracy model (dna_r10.4.1_e8.2_400bps_sup@v5.0.0) and the modified base call model 5mC_5hmC@v1. Base modifications calls were extracted for different sequence contexts (CG, CA, CAC) with modkit v0.3.1 (https://github.com/nanoporetech/modkit), setting the threshold for base modification calling to 0.8 (--filter-threshold C:0.8). Base modification calls for autosomes were aggregated to calculate the overall modification level for each sample. Samples with insufficient sequencing coverage, i.e. fewer than 70,000 measured cytosines per context (CG, CAC) equivalent to about 0.003X coverage, were discarded from the analysis.

## Haematopoietic reconstitution and competitive bone marrow transplantation assays

Recipient mice on a C57BL/6-Ly5.1/Ly5.2 (CD45.1$^+$CD45.2$^+$) genetic background were lethally irradiated with 2 doses of 5.5 Gy, 2 h apart. They were then injected intra-venously (i.v.) with foetal liver cells from E13.5 embryos or whole bone marrow, both rich sources of haematopoietic stem and progenitor cells (HSPCs), by intravenous (i.v.) tail injection. Recipient mice received a maximum of $2 \times 10^6$ cells in a maximum volume of 200 μL of PBS. Following successful transplantation, mice were monitored weekly by animal technicians for signs of illness (as above).

Bone marrow was harvested from both femora of donor mice. The bone marrow was gently flushed and cells counted. Competitor bone marrow derived from *C57BL/6/Ly5.1J* (CD45.1$^+$) mice was pooled. The CD45.2$^+$ bone marrow was harvested from either control WT mice or *Dnmt3a*$^{R878H/+}$ mice. The CD45.2$^+$ cells were not pooled to maintain separate biological replicates. Competitor (CD45.1$^+$) and test (CD45.2$^+$) bone marrow cells were mixed in PBS at a 1:1 ratio.

For serial competitive bone marrow reconstitution, recipient mice were left to age for 3 months following transplantation. At the end of the primary endpoint, bone marrow was harvested and cells counted. Bone marrow cells from each technical replicate were pooled in equal ratios to minimise technical variation. A total of $2 \times 10^6$ bone marrow cells from each mixture were injected i.v. into each lethally γ-irradiated secondary Ly5.1 recipient mouse. Blood, spleen, thymus and bone marrow were also collected for analysis by flow cytometry and histology.

## Flow cytometric analysis and cell sorting

Single-cell suspensions were generated for each haematopoietic organ of interest, before cells were stained with a cocktail of

antibodies against cell surface markers in the presence of anti-FC γ R block (1:10, made in-house) in wash buffer (10% foetal bovine serum (SAFC) in PBS (Gibco)), protected from light on ice for 30 min. Immediately prior to analysis by flow cytometry or before fluorescence-activated cell sorting (FACS), live cells were treated with a viability marker PI (2 µg/mL, Sigma-Aldrich), DAPI (1 µg/mL, BioLegend), Fluorogold (1 µg/mL, AAT Bioquest), or TO-PRO-3 Stain (1 µM, ThermoFisher Scientific). Flow cytometry was performed on either an LSRII, Fortessa1, or Fortessa X20 (BD Biosciences) or Aurora (Cytek) machine. Cell sorting was performed by staff at the WEHI FACS facility using a FACSAria III or FACSAria Fusion 9 machine (BD Biosciences). All data were analysed using FlowJo v10 (BD Biosciences). The cell types identified are defined in Table EV1, and all antibodies used for surface marker staining are listed in Reagents Table.

## RNA-seq analysis

Four-week-old female *Dnmt3a^{R878H/+}* mice or WT control littermates received a single low dose of γ-radiation (2 Gy) or were left untreated (baseline). 2 h after γ-irradiation, mice were sacrificed, and their bone marrow used to generate single-cell suspension, which were then stained with biotinylated antibodies against mature lineage surface markers. Magnetic selection using MagniSort Streptavidin Negative Selection Beads (Thermo Fisher, MSNB-6002-74) was used to remove mature lymphoid, myeloid and erythroid cells, following the manufacturer's protocol. The lineage-negative enriched bone marrow cells were then stained with fluorochrome conjugated antibodies against c-KIT and SCA-1 to identify HSPCs, as well as fluorochrome conjugated antibodies against the mature lineage surface markers (as above) to detect any non-depleted mature lineage cells. Cells were finally stained with PI (2 µg/mL) and sorted (staining and sorting performed as above). The cell types identified are defined in Table EV1, and all antibodies used for surface marker staining are listed in Reagents Table.

RNA from LSK cells was extracted using an miRNeasy kit (Qiagen, 217084) with optional on-column DNase digestion (Qiagen, 79254). Libraries were prepared using the Nextera XT DNA Library prep kit (Illumina) according to the manufacturer's instructions.

Thirteen RNA-seq libraries were included in the analysis. Adapter sequences were trimmed using Trim Galore v0.4.5. Sequencing reads were aligned to the mouse genome, mm10, using the Rsubread package v2.0.1 (Liao et al, 2019). Over 92.3% of reads mapped to the reference genome for each library. Successfully mapped reads were summarised into gene-level counts using featureCounts from Rsubread, with between 74.9% and 83.3% of reads assigned to genes. Genes were identified using GENCODE annotation (Frankish et al, 2018) for the mouse genome vM23. Differential expression (DE) analysis was undertaken using the limma (Ritchie et al, 2015) and edgeR (Robinson et al, 2009) packages v3.44.3 and v3.30.3, respectively. The DE analysis was restricted to protein-coding genes only. Haemoglobin genes as well as lowly expressed genes were filtered out. This resulted in 13,607 genes being retained for downstream analysis. Compositional differences between the libraries were normalised using the trimmed mean of M-values (TMM) method (Robinson and Oshlack, 2010). Read counts were converted to log2 counts per million (logCPM) and differential gene expression between groups of samples was assessed using linear models and robust empirical Bayes moderated t-statistics with a trended prior variance

while incorporating three surrogate variables for batch correction. *P*-values were adjusted to control for the false discovery rate (FDR) at below 10% using the Benjamini and Hochberg method. Analyses of the Gene Ontology (GO) terms and KEGG pathways were performed using limma's goana and kegga functions, respectively. Rotation gene set tests were performed and Gene Set Enrichment Analysis (GSEA) was used to generate the *p53* pathway enrichment plot. Additional barcode plots were generated using limma's roast (Ritchie et al, 2015) and barcodeplot functions, respectively, to test for the enrichment of genes associated with the *p53* pathway. Gene tracks were created in Integrated Genomic Viewer open access package from the Broad Institute (Robinson et al, 2011).

## Mito stress test

To determine the oxygen consumption rate (OCR) in cells, an XFe96 Extracellular Flux Analyzer was used. The assay was performed according to the manufacturer's instructions using the Seahorse XF Cell Mito Stress Test Kit User Guide (Agilent Technologies, 103016-400). $8.0 \times 10^5$ B lymphocytes or $6.0 \times 10^5$ myeloid cells per well were plated on a Seahorse Plate for this assay. Baseline respiration was measured in XF RPMI medium (supplemented with 1 mM pyruvate, 2 mM glutamine, and 10 mM glucose). Oxygen consumption was further measured in differing metabolic conditions, following the addition of 3 µM Oligomycin, 2 µM FCCP and 2 µM Antimycin A/1 µM Rotenone.

## Statistical analysis and preparation of figures

Prism software (Graphpad) was used to generate graphs, survival curves, and to analyse data unless otherwise indicated. Statistical methods used are referenced in the figure legends of relevant data. Some figures were drawn using assets from Biorender.com.

# Data availability

Low coverage Nanopore DNA sequencing data were deposited into the European Nucleotide Archive under accession number PRJEB78856 and are available at the following URL: https://www.ebi.ac.uk/ena/browser/view/PRJEB78856. RNA sequencing data were deposited into the Gene Expression Omnibus under accession number GSE275532 and are available at the following URL: https://www.ncbi.nlm.nih.gov/geo/query/acc.cgi?&acc=GSE275532. Source data for the main figures were deposited onto BioStudies under accession number S-BSST1870 and are available at the following URL: https://www.ebi.ac.uk/biostudies/studies/S-BSST1870?key=68e00733-cfcf-4d80-b954-c86f4344e441.

The source data of this paper are collected in the following database record: biostudies:S-SCDT-10_1038-S44319-025-00450-4.

# Peer review information

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

## Acknowledgements

We thank the entire Herold, Strasser and Kelly laboratories, Animal technicians, the FACS laboratory, the Genomics lab and all other WEHI support services. EL was funded by a joint PhD scholarship from the Australian Rotary Health District 9650 and The Walter and Eliza Hall Institute. AC was funded by a Alfred Hughes PhD Scholarship (WEHI). Grants for AS: Investigator Grant #1020363, NHMRC Program Grant #101671; Grants for MJH: NHMRC Investigator Grant 2017971, NHMRC Project Grants GNT1159658, GNT1186575, GNT1145728 and GNT1143105, Grant-in-Aid (CCV). The generation of the Dnmt3a^{R878H/+} mice and Puma-tdTomato reporter mice used in this study was supported by Phenomics Australia and the Australian Government through the National Collaborative Research Infrastructure Strategy (NCRIS) program. This work was supported by operational infrastructure grants through the Australian Government NHMRC IRIISS and the Victorian State Government Operational Infrastructure Support.

## Author contributions

**Erin M Lawrence**: Investigation. **Amali Cooray**: Investigation. **Andrew J Kueh**: Resources. **Martin Pal**: Resources. **Lin Tai**: Resources. **Alexandra L Garnham**: Formal analysis. **Connie S N Li-Wai-Suen**: Formal analysis. **Hannah Vanyai**:

Investigation. **Quentin Gouil**: Formal analysis. **James Lancaster**: Formal analysis. **Sylvie Callegari**: Investigation. **Lauren Whelan**: Investigation. **Elizabeth Lieschke**: Resources. **Annabella Thomas**: Resources. **Andreas Strasser**: Supervision. **Yang Liao**: Formal analysis. **Wei Shi**: Supervision. **Andrew H Wei**: Supervision. **Marco J Herold**: Conceptualization; Funding acquisition.

Source data underlying figure panels in this paper may have individual authorship assigned. Where available, figure panel/source data authorship is listed in the following database record: biostudies:S-SCDT-10_1038-S44319-025-00450-4.

## Disclosure and competing interests statement

The authors declare no competing interests.

# Expanded View Figures

**Figure EV1. Phenotyping of the *Dnmt3a^{R878H/+}* mutant mice.**

(A) Representative flow cytometry plots of spleen cells from *Dnmt3a^{R878H/+}* mutant mice and WT littermates. Leukocytes were gated by FSC-A and SSC-A, then cell doublets were gated out by FSC-H and FSC-A (not shown). Live cells were gates as being negative for the dead cell marker used, i.e. PI, DAPI, or Fluorogold. Cell subsets were identified based on their expression of cell type specific surface markers: CD4 + T cells = TCRβ + /CD4 + /CD8-; CD8 + T cells = TCRβ + /CD4-/CD8+); transitional B cells = B220 + /IgD$_{low}$/IgM$_{high}$; mature B cells = B220 + /IgD + /IgM$_{mid}$; granulocytes = TCRβ-/B220-/GR1 + /MAC1+; and macrophages = TCRβ-/B220-/GR1-/MAC1+. (B) Percentages of the indicated haematopoietic cell subsets determined by flow cytometry in the blood and spleen of mice of the indicated genotype. (C–H) Blood cell composition in mice with γ-irradiation induced thymic T cell lymphoma: lymphocytes (C), neutrophils (D), monocytes (E), eosinophils (F), basophils (G), and large unstained cells (LUCs) (H) determined by ADVIA analysis. (I) Flow cytometry plots of TCRβ and B220 expression on spleen cells from thymic T lymphoma burdened *Dnmt3a^{R878H/+}* mutant mice and WT littermates collected at ethical endpoint. Leukocytes were gated by FSC-A and SSC-A, then cell doublets were gated out by FSC-H and FSC-A (not shown). Live cells were gated as being negative for the dead cell marker used, i.e. PI, DAPI, or Fluorogold. (J) Flow cytometry plots of TCRβ and B220 expression in thymus samples from thymic T lymphoma burdened *Dnmt3a^{R878H/+}* mice and WT littermates collected at the ethical endpoint. Leukocytes were gated by FSC-A and SSC-A, then cell doublets were gated out by FSC-H and FSC-A (not shown). Live cells were gates as being negative for dead cell markers PI, DAPI, or Fluorogold. (K) Graphical representations of the proportions of TCRβ and B220 expression in thymus of thymic T cell lymphoma burdened *Dnmt3a^{R878H/+}* mice and WT littermates represented in (J). Data information: Error bars are the mean (±SEM) of (B) $n = 4$ *Dnmt3a^{R878H/+}* and $n = 5$ *Dnmt3a^{+/+}* (blood), $n = 6$ *Dnmt3a^{R878H/+}* and $n = 6$ *Dnmt3a^{+/+}* (spleen), (C–H) $n = 7$ *Dnmt3a^{R878H/+}* and $n = 8$ *Dnmt3a^{+/+}*, and (K) $n = 7$ *Dnmt3a^{R878H/+}* and $n = 5$ *Dnmt3a^{+/+}* independent biological repeats. Statistical significance was assessed using Prism 10 software by t tests. ns, not significant; $p > 0.05$, *$p < 0.05$.

▶

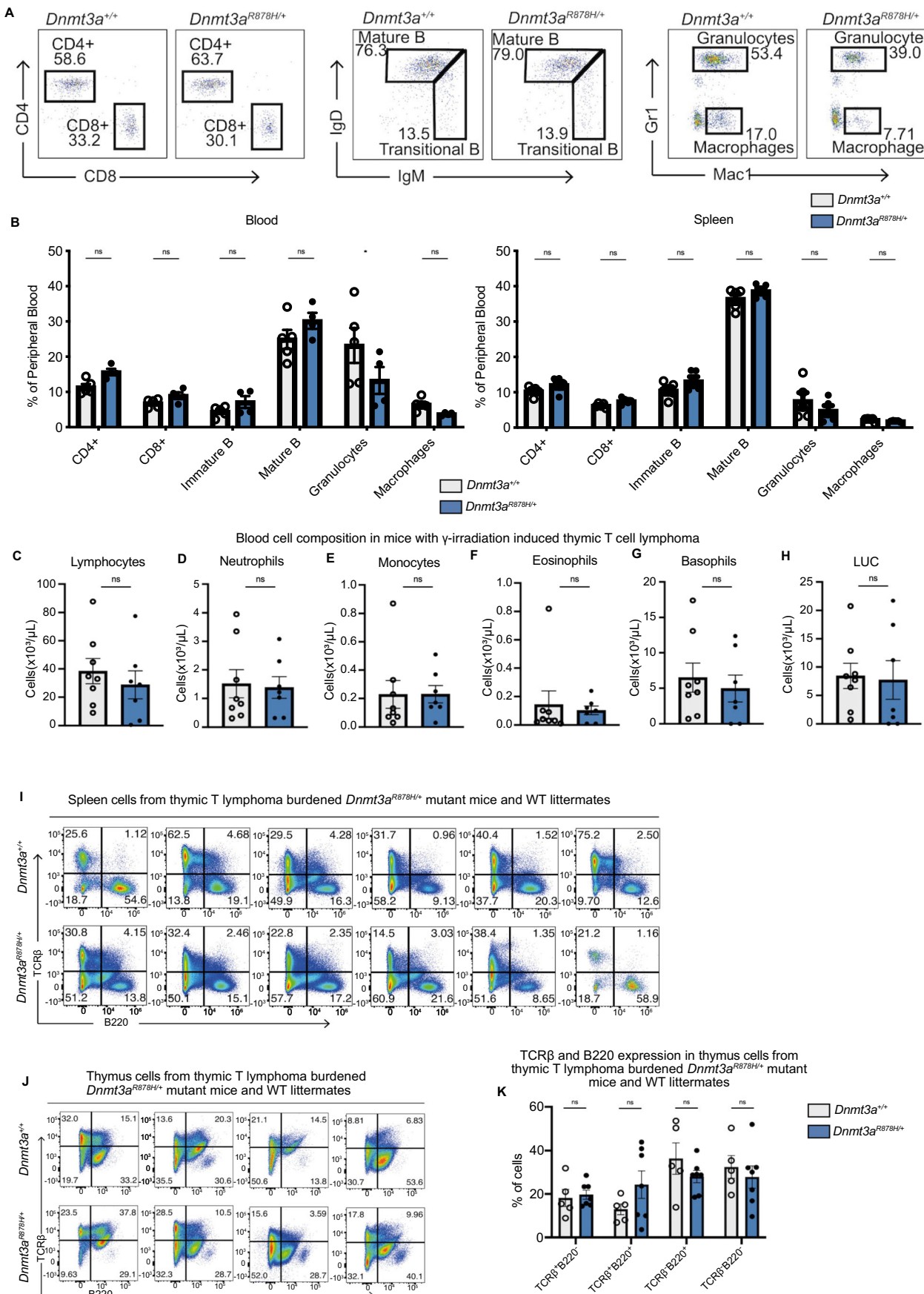

**B** Blood cell composition in mice with γ-irradiation induced thymic T cell lymphoma

**I** Spleen cells from thymic T lymphoma burdened *Dnmt3a^{R878H/+}* mutant mice and WT littermates

**J** Thymus cells from thymic T lymphoma burdened *Dnmt3a^{R878H/+}* mutant mice and WT littermates

TCRβ and B220 expression in thymus cells from thymic T lymphoma burdened *Dnmt3a^{R878H/+}* mutant mice and WT littermates

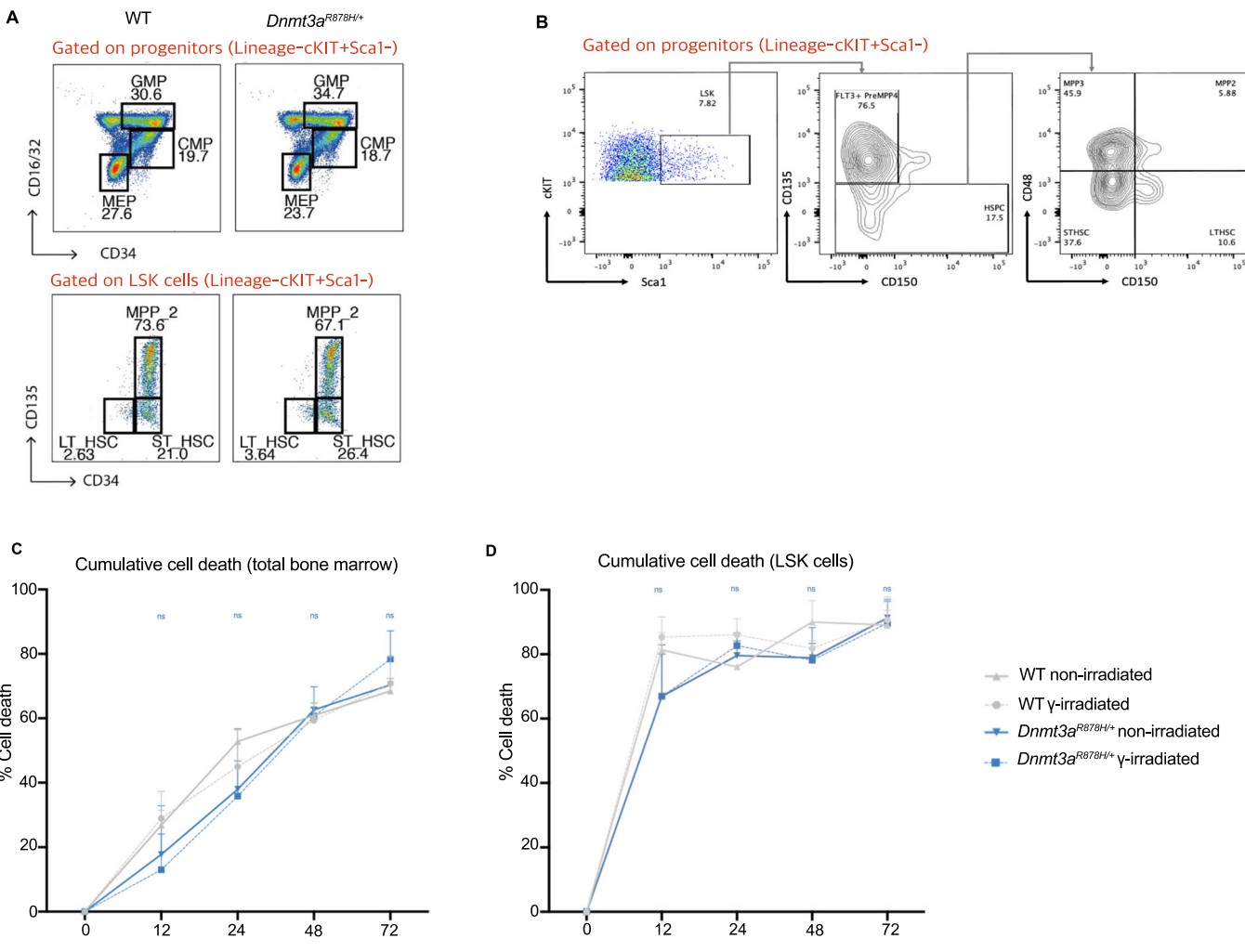

**Figure EV2. Flow cytometry gating strategies and ex vivo γ-irradiation induced cell death analysis.**

(A) Flow cytometry gating strategy to identify HSPCs amongst total bone marrow cells in mice. Granulocyte myeloid progenitors (GMP), common myeloid progenitors (CMP), megakaryocyte erythroid progenitors (MEP), multipotent progenitor cells (MPP), long-term haematopoietic stem cells (LT-HSC), short-term haematopoietic stem cells (ST-HSC). (B) Flow cytometric gating strategy to define the different cell subsets in the HSPC compartment when assessing γ-irradiation-induced cell death. Long-term haematopoietic stem cells (LT-HSC), short-term haematopoietic stem cells (ST-HSC), megakaryocyte-biased multipotent progenitor subset (MPP2), and myeloid-biased multipotent progenitor subset (MPP3) multipotent progenitor cells 3. (C, D) Representations of cumulative cell death from flow cytometric analyses of (C) total cells and (D) LSK cells from the bone marrow of $Dnmt3a^{R878H/+}$ mutant mice and WT littermates, following 1.5 Gy γ-irradiation or untreated controls. Percentage of cell death is presented relative to time point 0. Data information: (C, D) Error bars are the mean (±SEM). The experiment was performed with three independent biological replicates ($n = 3$ $Dnmt3a^{R878H/+}$ and $n = 3$ $Dnmt3a^{+/+}$ γ-irradiated mice, and $n = 3$ $Dnmt3a^{R878H/+}$ and $n = 3$ $Dnmt3a^{+/+}$ non-irradiated mice) with three technical replicates, resulting in a total of 9 mice per genotype and treatment. Statistical significance was assessed using Prism 10 software by 2-way ANOVA with Šídák's multiple comparisons test; ns, not significant; $p > 0.05$.

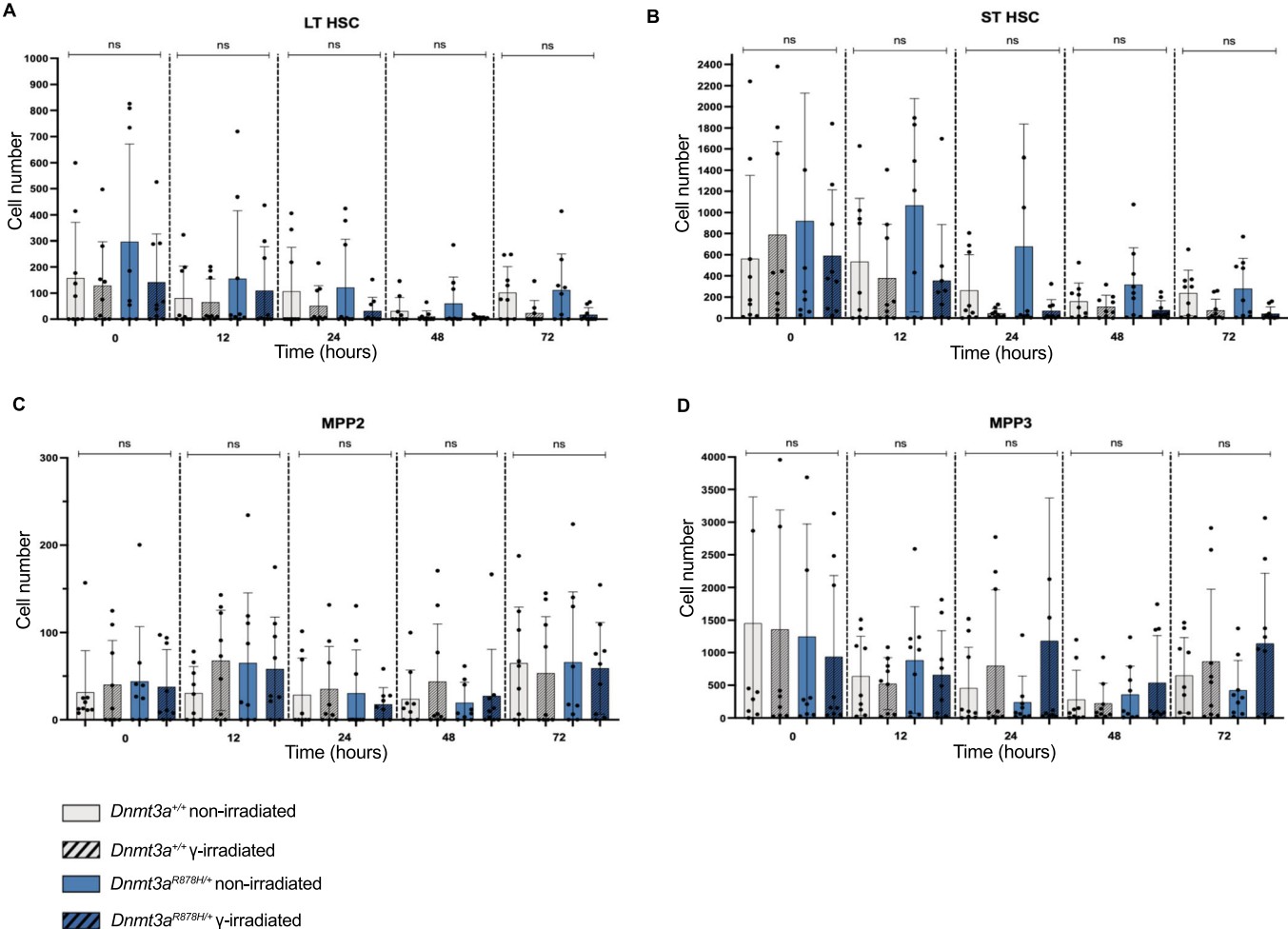

**Figure EV3.  Ex vivo irradiation induced cell death analysis.**

Bone marrow was harvested from 8–12-week-old mice and left untreated, or given 1.5 Gy γ-irradiation immediately after harvesting. These cells were placed in culture and measured by flow cytometry at baseline, 12-, 24-, 48- and 72-h timepoints. (A–D) Flow cytometric analyses of the HSPC compartment were used to determine the cell numbers at each time point for long-term haematopoietic stem cells (LT-HSC) (A), short-term haematopoietic stem cells (ST-HSC) (B), megakaryocyte-biased multipotent progenitor subset (MPP2) (C), and myeloid-biased multipotent progenitor subset (MPP3) (D). For each condition, $n = 9$ mice (3 mice per treatment and genotype across 3 independent experiments). Data are presented as mean $+/-$ SEM. Data information: Error bars are the mean (±SEM). The experiment was performed with three independent biological replicates ($n = 3$ $Dnmt3a^{R878H/+}$ and $n = 3$ $Dnmt3a^{+/+}$ γ-irradiated mice, and $n = 3$ $Dnmt3a^{R878H/+}$ and $n = 3$ $Dnmt3a^{+/+}$ non-irradiated mice) with three technical replicates, resulting in a total of 9 mice per genotype and treatment. Statistical significance was assessed using Prism 10 software by 2-way ANOVA with Tukey's multiple comparisons test; ns, not significant; $p > 0.05$.

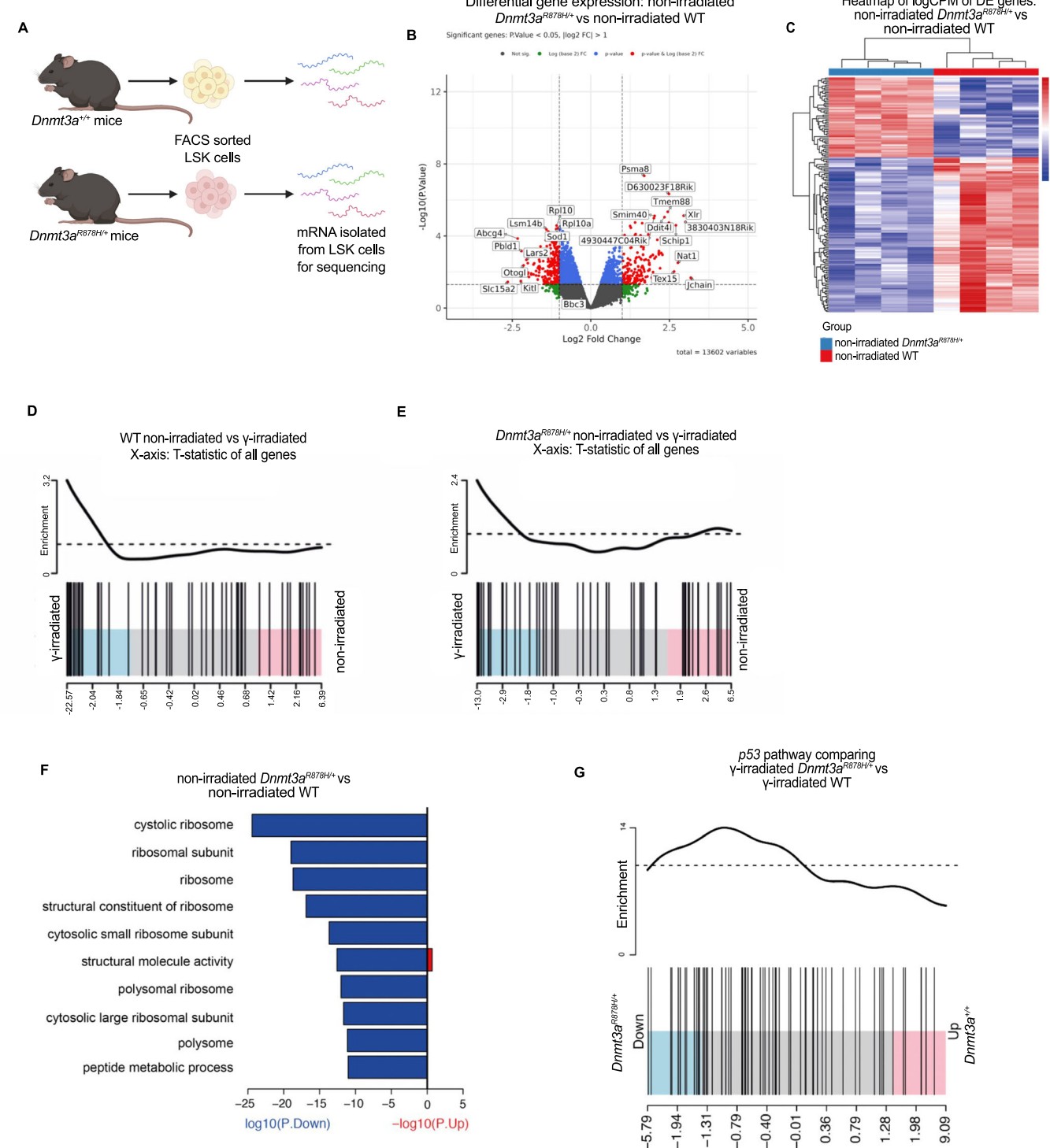

**B** Differential gene expression: non-irradiated *Dnmt3a^R878H/+* vs non-irradiated WT

**C** Heatmap of logCPM of DE genes: non-irradiated *Dnmt3a^R878H/+* vs non-irradiated WT

**D** WT non-irradiated vs γ-irradiated X-axis: T-statistic of all genes

**E** *Dnmt3a^R878H/+* non-irradiated vs γ-irradiated X-axis: T-statistic of all genes

**F** non-irradiated *Dnmt3a^R878H/+* vs non-irradiated WT

**G** *p53* pathway comparing γ-irradiated *Dnmt3a^R878H/+* vs γ-irradiated WT

◀ **Figure EV4.  RNAseq of LSK cells from untreated and γ-irradiated *Dnmt3a^(R878H/+)* mice and WT littermates.**

(A) Schematic depicting the experimental workflow whereby bone marrow was harvested from untreated age-matched female *Dnmt3a^(R878H/+)* mice and WT littermates. LSK cells were isolated by FACs from the bone marrow, and their mRNA extracted for RNA-seq. (B) Volcano plot depicting differences in gene expression between LSK cells harvested from non-irradiated *Dnmt3a^(R878H/+)* mice cells *vs* non-irradiated WT control mice where genes in red are significantly differentially expressed and ranked by *p*-value and log fold change. *Puma/Bbc3* has been highlighted. (C) Hierarchical clustering heatmap depicting differentially expressed genes between non-irradiated *Dnmt3a^(R878H/+)* and WT LSK cells. Red and blue squares indicate genes with high or low gene expression levels, respectively. (D, E) Barcode plots showing enrichment of genes in the p53 signalling pathway (KEGG pathway mmu04115) in non-irradiated WT LSK cells vs γ-irradiated WT LSK cells (D) and non-irradiated *Dnmt3a^(R878H/+)* LSK cells vs γ-irradiated *Dnmt3a^(R878H/+)* LSK cells (E). (F) Gene ontology analysis reveals the top 10 downregulated pathways in untreated *Dnmt3a^(R878H/+)* LSK cells compared to untreated WT LSK cells. (G) Barcode plot showing enrichment of genes in the p53 signalling pathway (KEGG pathway mmu04115) in γ-irradiated *Dnmt3a^(R878H/+)* LSK cells compared to γ-irradiated WT LSK cells. Data information: (B, F) $n = 4$ *Dnmt3a^(R878H/+)* and $n = 4$ *Dnmt3a^(+/+)* independent biological repeats. Statistical significance criteria is $p < 0.05$, $\log_2 FC > 1$. Statistical significance was determined using the limma-trend method with empirical Bayes moderation after fitting a linear model to log-transformed counts per million (logCPM). Surrogate variable analysis (SVA) was incorporated to adjust for hidden sources of variation. GO enrichment analysis was conducted using goana.

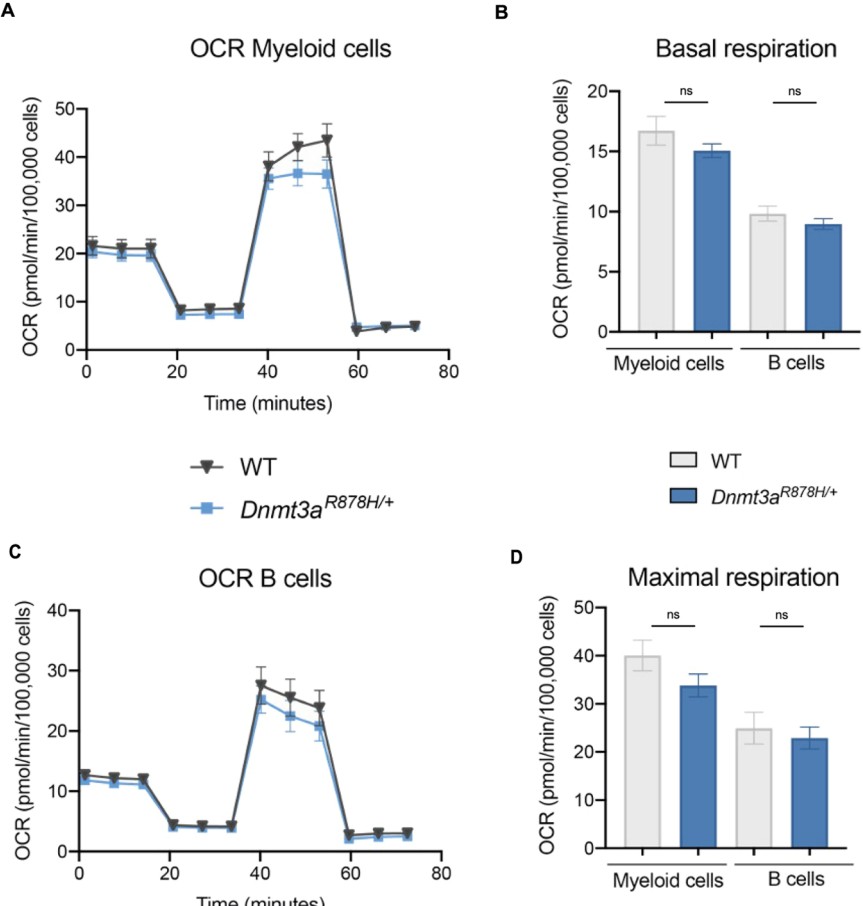

**Figure EV5. Oxygen consumption rates of bone marrow cells from *Dnmt3a^R878H/+* mutant mice and their WT littermates.**

(**A**) Oxygen consumption rate (OCR) profiles of bone marrow-derived myeloid cells from *Dnmt3a^R878H/+* or *Dnmt3a^+/+* mice. (**B**) OCR profile of B lymphoid cells from the bone marrow of *Dnmt3a^R878H/+* or *Dnmt3a^+/+* mice. (**C**) Basal respiration rates for myeloid cells and B lymphoid cells from the bone marrow of *Dnmt3a^R878H/+* or *Dnmt3a^+/+* mice. (**D**) Maximal respiration rates for bone marrow-derived myeloid and B cells from *Dnmt3a^R878H/+* or *Dnmt3a^+/+* mice. Data information: All error bars are the mean (±SEM) of $n = 7$ *Dnmt3a^R878H/+* and $n = 5$ *Dnmt3a^+/+* independent biological repeats. Statistical significance was assessed using Prism 10 software by t tests. ns, not significant; $p > 0.05$.

A

Expression of p53 Pathway Genes

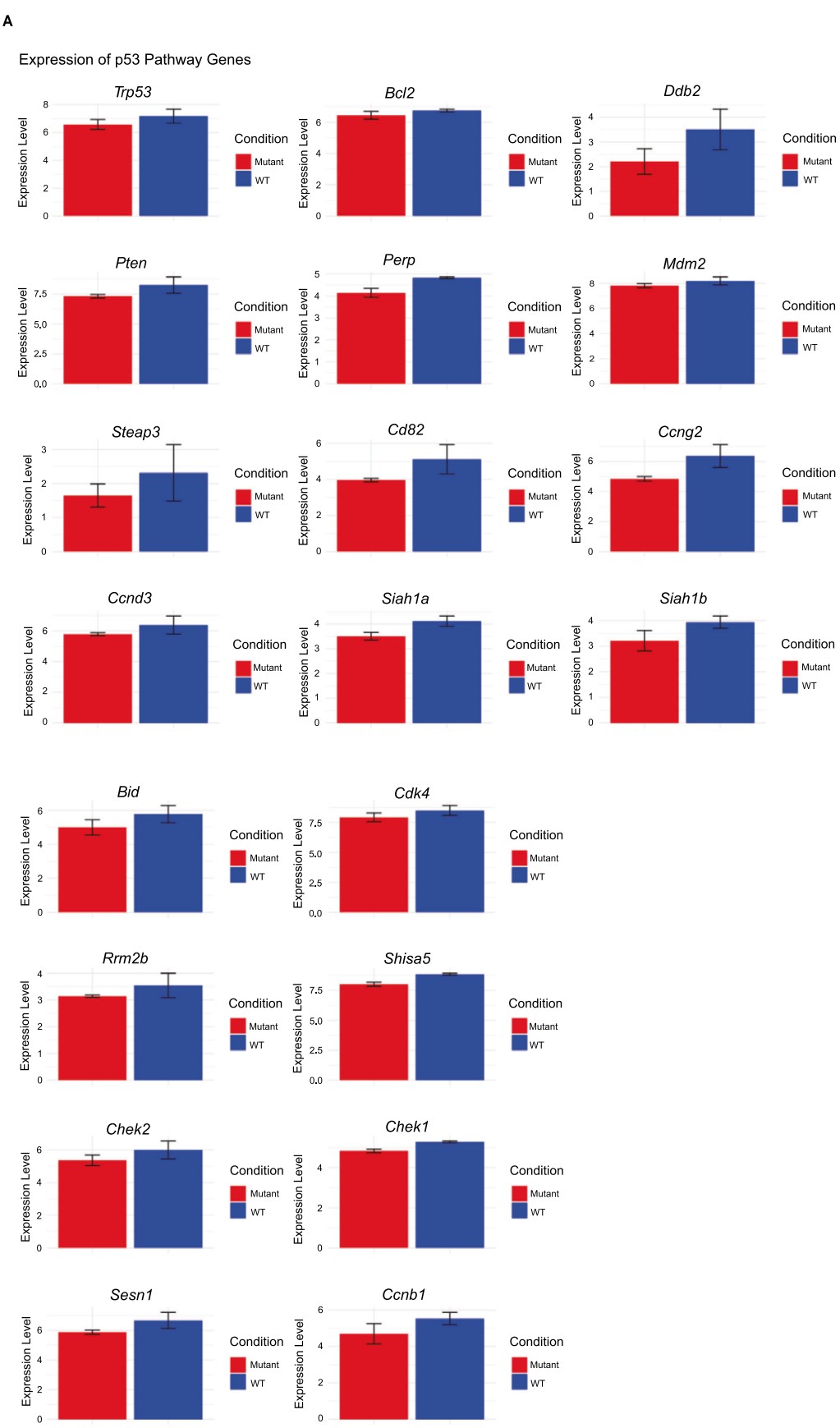

◀ **Figure EV6.  *p53* pathway genes in LSK cells from *Dnmt3a$^{R878H/+}$* mice and WT control littermates following 1.5 Gy γ-radiation.**

(A) RNA-seq analysis of LSK cells from *Dnmt3a$^{R878H/+}$* mice and WT control littermates following 2 Gy γ-radiation. Bar plots of the relative expression level of p53 pathway genes, where irradiated *Dnmt3a$^{R878H/+}$* (mutant) cells (red) show lower expression as compared to irradiated WT cells (blue). Data information: All error bars are the mean (±SEM) of $n = 3$ *Dnmt3a$^{R878H/+}$* and $n = 2$ *Dnmt3a$^{+/+}$* independent biological repeats. Statistical significance was assessed by t tests. ns, not significant; $p > 0.05$.

**Figure EV7. *Puma-tdTomato* reporter induction in thymocytes following γ-irradiation.**

Graphical representation *Puma-tdTomato* expression in the thymocytes of γ-irradiated *Dnmt3a^R878H/+^/Puma-tdTomato^KI/+^* and *Dnmt3a^+/+^/Puma-tdTomato^KI/+^*, and non-irradiated *Dnmt3a^R878H/+^/Puma-tdTomato^KI/+^* and *Dnmt3a^+/+^/Puma-tdTomato^KI/+^* mice. Data information: $n = 5$ γ-irradiated *Dnmt3a^R878H/+^/Puma-tdTomato^KI/+^*, $n = 5$ non-irradiated *Dnmt3a^R878H/+^/Puma-tdTomato^KI/+^*, $n = 5$ γ-irradiated *Dnmt3a^+/+^/Puma-tdTomato^KI/+^*, and $n = 5$ non-irradiated *Dnmt3a^+/+^/Puma-tdTomato^KI^* independent biological repeats. Error bars are the mean (±SEM). Statistical significance was assessed using Prism 10 software by 2-way ANOVA with Šídák's multiple comparisons test. ns, not significant; $p > 0.05$.

