## [Peer Review File · EMBO Reports]

Transcriptomic changes including p53 dysregulation prime DNMT3A mutant cells for transformation

Marco Herold, Erin Lawrence, Amali Cooray, Andrew Kueh, Martin Pal, Lin Tai, Alexandra Garnham, Connie Li-Wai-Suen, Hannah Vanyai, Quentin Gouil, James Lancaster, Sylvie Callegari, Lauren Whelan, Elizabeth Lieschke, Annabella Thomas, Andreas Strasser, Yang Liao, Wei Shi, and Andrew Wei

Corresponding author(s): Marco Herold (herold@wehi.edu.au)

Review Timeline:

Transfer Date:	22nd Nov 24
Editorial Decision:	22nd Nov 24
Revision Received:	16th Feb 25
Editorial Decision:	25th Mar 25
Revision Received:	2nd Apr 25
Accepted:	4th Apr 25

Editor: Achim Breiling

Transaction Report: This manuscript was transferred to EMBO reports following peer review at The EMBO Journal.

Dear Dr. Herold,

Thank you for transferring your manuscript to EMBO reports after our call to discuss the revision. As you know, the referees at The EMBO Journal had several concerns and suggestions to improve the manuscript, or to strengthen the data and the conclusions drawn. After our discussion, I would thus like to invite you to revise your manuscript as suggested, with the understanding that all concerns of the referees will be addressed in the revised manuscript or in a detailed p-b-p-response.

Acceptance of your manuscript will depend on a positive outcome of another round of review at EMBO reports, using the same referees.

It will also be necessary that during the revision you address all technical concerns, or points regarding the experimental designs, model systems used, or data presentation.

Revised manuscripts should be submitted within three months of a request for revision. Please contact me to discuss the revision if you have questions or comments regarding the revision, or should you need additional time.

1) a .docx formatted version of the final manuscript text (including legends for main figures, EV figures and tables), but without the figures included. Please make sure that changes are highlighted to be clearly visible. Figure legends should be compiled at the end of the manuscript text.

2) individual production quality figure files as .eps, .tif, .jpg (one file per figure), of main figures and EV figures. Please upload these as separate, individual files upon re-submission. Please make sure that all figure panels are called out separately and sequentially in the manuscript text

For more details please refer to our guide to authors:

See also our guide for figure preparation:

Moreover, please consult our guidelines for figure legend preparation:

4) a complete author checklist, which you can download from our author guidelines (<https://www.embopress.org/page/journal/14693178/authorguide>). Please insert page numbers in the checklist to indicate where the requested information can be found in the manuscript. The completed author checklist will also be part of the RPF.

5) that primary datasets produced in this study (e.g. RNA-seq, ChIP-seq and array data) are deposited in an appropriate public

database. This is now mandatory (like the COI statement). If no primary datasets have been deposited in any database, please state this in this section (e.g. 'No primary datasets have been generated and deposited').

The accession numbers and database should be listed in a formal "Data Availability " section (placed after Materials & Methods) that follows the model below. Please note that the Data Availability Section is restricted to new primary data that are part of this study.

Data availability

8) Regarding data quantification and statistics, please make sure that the number "n" for how many independent experiments were performed, their nature (biological versus technical replicates), the bars and error bars (e.g. SEM, SD) and the test used to calculate p-values is indicated in the respective figure legends (also for potential EV figures and all those in the final Appendix). Please also check that all the p-values are explained in the legend, and that these fit to those shown in the figure. Please provide statistical testing where applicable. Please avoid the phrase 'independent experiment', but clearly state if these were biological or technical replicates. Please also indicate (e.g. with n.s.) if testing was performed, but the differences are not significant. In case n=2, please show the data as separate datapoints without error bars and statistics.

See also:

<http://www.embopress.org/page/journal/14693178/authorguide#statisticalanalysis>

If n<5, please show single datapoints for diagrams. Please add to each legend (main, EV figures, Appendix, where applicable) a 'Data Information' section explaining the statistics used or providing information regarding replicates and scales. See: <https://www.embopress.org/page/journal/14693178/authorguide#figureformat>

9) Please add scale bars of similar style and thickness to any microscopic images, using clearly visible black or white bars (depending on the background). Please place these in the lower right corner of the images themselves. Please do not write on or near the bars in the image but define the size in the respective figure legend.

10) Please note our reference format:

11) We updated our journal's competing interests policy in January 2022 and request authors to consider both actual and perceived competing interests. Please review the policy <https://www.embopress.org/competing-interests> and add a statement declaring your competing interests. Please name that section 'Disclosure and Competing Interests Statement' and add it after the author contributions section.

12) Please order the sections like this using these names:

Title page - Abstract - Keywords - Introduction - Results - Discussion - Methods - Data availability section (DAS) -

Acknowledgements (including funding information) - Disclosure and Competing Interests Statement - References - Figure

legends - Expanded View Figure legends

13) Please provide the abstract written in present tense throughout.

14) Please make sure that all the funding information is also entered into the online submission system and is complete and similar to the one in the manuscript text file (in the Acknowledgements).

15) We now use CRediT to specify the contributions of each author in the journal submission system. CRediT replaces the author contribution section. Please use the free text box to provide more detailed descriptions. Thus, please do NOT provide your final manuscript text file with an author contributions section. See also guide to authors: <https://www.embopress.org/page/journal/14693178/authorguide#authorshippinguidelines>

16) All materials and methods used need to be described in the main text using our 'Structured Methods' format, which is required for all research articles. According to this format, the Methods section should include a Reagents and Tools Table (listing key reagents, experimental models, software, and relevant equipment and including their sources and relevant identifiers), uploaded as separate file, followed by a Methods section in which we encourage the authors to describe their methods using a step-by-step protocol format with bullet points, to facilitate the adoption of the methodologies across labs. More information on how to adhere to this format as well as downloadable templates (.doc or .xls) for the Reagents and Tools Table can be found in our author guidelines (section 'Structured Methods'):

I look forward to seeing a revised version of your manuscript when it is ready. Please let me know if you have questions or comments regarding the revision.

Kind regards,

Achim

Referee #1:

This study is focused on the role of DNMT3A mutations in hematologic malignancies. The authors have generated a new mouse model which has a germline knock-in (R878H) that corresponds to a common human mutation R882H. They report that these mice have an increased incidence of radiation-induced thymic lymphomas with hematopoietic stem cells (LSK cells) from *Dnmt3a*^{R878/+} mice having an advantage over wild-type cells in competition assays. RNA-seq analysis of irradiated mutant LSK cells showed a downregulation of p53 target genes with reduced *Bbc3* (*Puma*) expression being demonstrated with a mouse strain containing a *tdTomato* reporter for its expression. The authors propose that DNMT3A mutation enhances self-renewal and transformation via effects on the p53-dependent transcriptome in LSK cells.

DNMT3A mutations have been clearly implicated in human hematologic malignancies. Elucidating the molecular mechanisms that contribute to their transforming activity is thus a highly significant area of study. The authors have generated a knock-in mouse which contains a highly relevant DNMT3a mutation, and this is an important contribution. However, the current manuscript is lacking in sufficient mechanistic insights to warrant publication in *The EMBO J*. Specific concerns are as follows:

1. The authors propose that alterations in the p53 transcriptome are important for the effects of the DNMT3A mutation. This

needs to be tested directly by showing that the gene signature that is identified is in fact p53-dependent. Reliance on gene set enrichment is insufficient. Use of a p53-null mouse or p53 knock-down is essential to make this conclusion.

2. Attention is drawn to the role of Puma in the observed outcomes. This aspect of the study needs to be substantially bolstered. Confirmation of the effects on Puma expression that are seen with the reporter need to be made by directly examining mRNA expression of Puma by RT-PCR and confirming that its transcriptional regulation is p53-dependent. The authors present data with effects of the DNMT3A mutation on global methylation levels and imply this is responsible for the effect on Puma. Methylation status on the gene encoding Puma is necessary to demonstrate this.

3. Most importantly, the Puma-dependence of the effects of the DNMT3A mutation need to be directly demonstrated. The reporter strain that is used has a knock-in of tdTomato making the Puma allele effectively null. The authors should use this mouse strain in the homozygous state to show that effects on thymic lymphoma development and LSK cell function are indeed Puma-dependent.

4. Puma has a well-characterized role in apoptosis. The observed phenotype appears to be associated with enhanced proliferation of LSK cells. The authors need to address how reduced expression of Puma contributes to this behavior of the LSK cells. Figure 4 is uninformative as there appears to be no statistical differences between wild-type and mutant.

5. The authors should also discuss further the gene set that is considered to be p53-dependent. What other well-known p53 targets besides Puma were found? Can the regulation of a subset of these be confirmed by RT-qPCR? And as noted above, is their expression indeed p53-dependent?

Additional points:

1. Details for the RNA-seq analysis are lacking or confusing, especially Figure 5 and the accompanying Supplemental Figure 4. A Volcano plot may be a better way to show the differential gene expression. It should be clearly shown what the cut-offs (fold-change and Pvalue) were used to identify differentially expressed genes. This is lacking for the current Figure 5B. How did the authors define the red and blue dots? Where is Bbc3 (Puma) in this plot?

2. Details for the gene ontology analysis showing effects on the p53 transcriptome are lacking. The figure legend for Supplemental Figure 4, especially parts D, E, and F need more information to clearly support the claims being made in the text.

3. 2Gy treatment is used for the RNA-seq studies whereas 5Gy is used for the tdTomato studies. The difference in radiation dose should be explained. Is Puma expression affected with 2Gy and is that found in the RNA-seq dataset (see above)?

4. The gene that encodes the Puma protein is called Bbc3. The author should take care to use the appropriate designation to avoid any confusion.

Referee #2:

The manuscript by Lawrence et al. describes the role of the methyltransferase DNMT3A in cell transformation. Using a heterozygous mouse model system for a hotspot DNMT3A mutation in humans R882H (R878H in mice) the authors performed a systematic analysis on the role of this mutation in the initiation of hematologic malignancies. The authors found that the R878H mutation causes a global reduction in DNA methylation across all tissues. This was accompanied by a sensitivity in the development of g-irradiation dependent thymic lymphoma and an advantage for self-renewal over differentiation. Subsequent RNAseq analysis showed that the R878H heterozygous animals exposed to low doses of g-irradiation had reduced expression of p53-regulated pathways. Notably, expression of PUMA, a well-described p53-dependent apoptosis induced gene, was downregulated in the mutant animals.

The study is very well presented and the data are of very high quality supporting the key interpretations of the authors.

While I fully acknowledge the in depth and systematic phenotypic characterization of the DNMT3A R878H animals, which is important, I feel that study currently suffers from lack of sufficient mechanistic details.

In particular, the defects in the p53-regulated pathways are interesting but:

-Is the effect related to the general observed defect in DNA methylation? DNMT3A is known to interact with p53 and control the expression of p53-dependent genes such as p21, but possibly in methyltransferase activity independent manner (PMID: 31640986). Paradoxically to the presented study, in some cases, DNMT3A repressed p53-dependent gene expression.

-Does the mutation affect p53 stability, post-translational modifications?

-Are the observed defects in PUMA expression p53 dependent?

-Would the observed defects in PUMA expression the main driver for the observed phenotypes? would potential defects in cell cycle regulating factors such as p21 are mediating the observed defects in self-renewal and/or differentiation?

Re: Submission of Manuscript ID: EMBOR-2024-60834

Point-by-point response to the referees' concerns

Referee #1:

- 1. The authors propose that alterations in the p53 transcriptome are important for the effects of the DNMT3A mutation. This needs to be tested directly by showing that the gene signature that is identified is in fact p53-dependent. Reliance on gene set enrichment is insufficient. Use of a p53-null mouse or p53 knock-down is essential to make this conclusion.***

We appreciate the concern from Referee #1 that reliance on gene set enrichment alone is insufficient. Our lab has previously generated *Dnmt3a*^{R878H/+}/*Trp53*^{+/-} mice, and we found that there was no significant difference in overall and tumor-free survival when the *Dnmt3a* mutation is combined with loss of one allele of *p53*. These data have now been included as Figure 5A in the revised manuscript.

Figure for referees not shown.

While we tried to generate crosses with homozygous *p53*-null mice on the *DNMT3a* mutant background, the breeding issues with *DNMT3a*^{+/-} female mice and ethical considerations made such a cross impossible.

Our results indicate that the *Dnmt3a*^{R878H/+} mutation does not cooperate with the loss of one allele of *Trp53* to accelerate tumour development. This is consistent with published literature. Indeed, existing sequencing datasets of human AML patients show that *DNMT3A* and *TP53* mutations tend not to co-occur within the same samples (Weinstein et al. 2013). This mutual exclusivity of these two mutations suggests that the genes operate in the same pathway and would therefore not be expected to cooperate in tumorigenesis. Hence, we propose that

the *Dnmt3a*^{R878H/+} mutation causes *p53* pathway dysregulation such that removing one allele of *Trp53* might not produce additive effects in tumorigenesis because the pathway is already dysregulated to the threshold needed for driving tumorigenesis.

In our previous manuscript submission, the *p53* pathway was shown using a barcode plot (previously Figure 5D), which visualises the distribution of *p53* pathway genes. An alternative gene set enrichment analysis tool, FGSEA (Korotkevich et al. 2021), was used to apply a more rigorous statistical approach to gene set testing, and a new enrichment plot was generated of the *p53* pathway in LSK cells isolated from γ -irradiated *Dnmt3a* mutant or γ -irradiated wild-type mice. This plot appears consistent with what was shown in the barcode plot, while also being easier to interpret. This new plot has been included in the revised manuscript as Figure 4D, while the barcode plot has been relocated to Expanded View Figure EV4G.

Figures for referees not shown.

To examine this finding further, the below box plot looks at the expression of just the *Trp53* gene in the γ -irradiated *Dnmt3a*^{R878H/+} samples vs the γ -irradiated wild-type samples. This shows that *Trp53* is subtly downregulated in the mutant group. These new data have been included in the manuscript as Figure 5B.

Figure for referees not shown.

In order to interrogate the *p53* pathway further, we identified the individual genes within it. We performed a t-test for each gene, comparing transcript enrichment by comparing γ -irradiated *Dnmt3a*^{R878H/+} LSK cells with γ -irradiated wild-type LSK cells. While none of the differences in gene expression were found to be statistically significant, many genes were identified to be downregulated in the γ -irradiated *Dnmt3a* mutant cells. These data have been included in the revised manuscript as Expanded view figure EV6.

Figure for referees not shown.

Although the differences in the expression of these genes are not statistically significant, the consistency of slight downregulation across multiple *p53* pathway genes might suggest that this effect is not random.

In summary, these new data strengthen our hypothesis that the *Dnmt3a* R878H mutation causes *p53* pathway dysregulation, which results impacts regulation of the genes acting downstream of *p53*.

2. Attention is drawn to the role of Puma in the observed outcomes. This aspect of the study needs to be substantially bolstered. Confirmation of the effects on Puma expression that are seen with the reporter need to be made by directly examining mRNA expression of Puma by RT-PCR and confirming that its transcriptional regulation is p53-dependent. The authors present data with effects of the DNMT3A mutation on global methylation

levels and imply this is responsible for the effect on Puma. Methylation status on the gene encoding Puma is necessary to demonstrate this.

We thank the Reviewer for this comment. The alteration of *Puma* reporter expression was used as a surrogate marker for dysregulation of *p53* function. When we initially analysed *Puma (Bbc3)* mRNA expression in a bulk RNA seq experiment comparing *Dnmt3a*^{R878H/+} LSK cells with wild-type LSK cells, both at baseline or after exposure to γ -irradiation we did not observe statistically significant differences (see below alignment of RNA peaks to the *Puma (Bbc3)* locus). However, using the *Puma* reporter mice allows a more sensitive way to analyse *Puma* gene locus activity as compared to bulk mRNA sequencing of LSK cells. We were able to show reduced *Puma* reporter induction in γ -irradiated bone marrow and spleen cells from *Dnmt3a*^{R878H/+} mice compared to γ -irradiated WT bone marrow and spleen cells. The *Puma* reporter was used here as a measure of transcriptional *p53* activity. Our findings provide evidence that it is the dysregulation of the *p53* pathway that causes the impairment in γ -radiation induced increase in the levels of *Puma/Bbc3* mRNA. We propose that the reduced expression of the *Puma* reporter in *Dnmt3a*^{R878H/+} cells is due to reduced expression of *p53*. Our manuscript has been revised to describe this more clearly. Please note that it is firmly established that *Puma* is a direct *p53* target gene from work of others (Yu et al. 2001; Nakano and Vousden 2001; Han et al. 2001).

3. Most importantly, the Puma-dependence of the effects of the DNMT3A mutation need to be directly demonstrated. The reporter strain that is used has a knock-in of tdTomato making the Puma allele effectively null. The authors should use this mouse strain in the homozygous state to show that effects on thymic lymphoma development and LSK cell function are indeed Puma-dependent.

The groups of Andreas Villunger (Labi et al. 2010) and Andreas Strasser (Michalak et al. 2010) have both shown independently that the absence of *Puma* prevents γ -radiation induced thymic T cell lymphoma development. Therefore, the homozygous *Puma* reporter cannot be used to observe the effect on the development of this lymphoma. Furthermore, we do not wish to suggest that

DNMT3a directly interacts with the *Puma* gene locus. Instead, we attribute the reduced *Puma* reporter expression in *Dnmt3a* mutant cells to a reduction in *p53* activity; i.e. the *Dnmt3a* mutation causes a reduction in the levels and activity of *p53*.

4. *Puma* has a well-characterized role in apoptosis. The observed phenotype appears to be associated with enhanced proliferation of LSK cells. The authors need to address how reduced expression of *Puma* contributes to this behavior of the LSK cells. Figure 4 is uninformative as there appears to be no statistical differences between wild-type and mutant.

We do not wish to suggest that it is solely the reduction of *Puma* that causes the increased proliferation of LSK cells in *Dnmt3a* mutant mice. Rather, the abnormally increased LSK cell proliferation is likely a consequence of the reduction of *p53* levels and activity (as a consequence of the *Dnmt3a* mutation) causing a minor reduction in many *p53* target genes, including *Puma*, rather than just one of its target genes. Possibly, a reduction in the levels and activity of *p53* will also reduce the expression of genes that encode negative regulators of cell cycling. Moreover, impact on indirect *p53* target genes could cause a decrease in genes that encode positive regulators of cell cycling. For instance, *Ccnd1* and *Ccnd2* (CyclinD1 and CyclinD2) are key regulators of cell cycle progression and survival, and both are frequently mutated in AML. The overexpression of these genes has previously been implicated in hematologic malignancy (Metcalf et al. 2010). Although not statistically significant, we found clear enrichment of mRNA of these genes in γ -irradiated *Dnmt3a* mutant LSK cells. We believe that the sum of these small effects on the expression of many genes (decrease in direct *p53* target genes and increase or decrease in indirect target genes) explains the increased proliferation of LSK cells observed in the *Dnmt3a* mutant mice. It is difficult to experimentally test this as a minor reduction in the levels of a large number of genes simultaneously is not feasible with current technology. We have expanded the discussion of these findings in our revised manuscript.

Figure for referees not shown.

Moreover, the small increase in *Mcl1* mRNA (anti-apoptotic protein), and reduced expression of DNA damage response genes (*Chek1*, *Chek2*, *Rad51*) in *Dnmt3a* mutant LSK cells, further indicate that it is the sum of many small effects that underlies the abnormally increased expansion of LSK cells in *Dnmt3a* mutant mice.

Figure for referees not shown.

The previous Figure 4 has been moved to become Expanded View Figure EV2C-D, and the above plots showing the expression levels of key genes in this gene set are now presented in Figure 5 in the revised manuscript.

5. The authors should also discuss further the gene set that is considered to be p53-dependent. What other well-known p53 targets besides Puma were found? Can the regulation of a subset of these be confirmed by RT-qPCR? And as noted above, is their expression indeed p53-dependent?

We believe that our response directly above has addressed this concern from Reviewer #1. While the individual genes of interest do not appear to be downregulated or upregulated (for some indirect *p53* target genes) to a point of statistical significance, the impact of the *Dnmt3a* R882H mutation on *Trp53* and downstream *p53* pathway genes has a cumulative affect of promoting cell survival and proliferation following DNA damage (see above).

Additional points:

1. Details for the RNA-seq analysis are lacking or confusing, especially Figure 5 and the accompanying Supplemental Figure 4. A Volcano plot may be a better way to show the differential gene expression. It should be clearly shown what the cut-offs (fold-change and Pvalue) were used to identify differentially expressed genes. This is lacking for the current Figure 5B. How did the authors define the red and blue dots? Where is *Bbc3* (*Puma*) in this plot)?

We thank the reviewer for their comment. To respond to this request, we have improved the presentation of the RNA-seq analysis. An updated Figure 5B has been generated, now showing a volcano plot annotated with the fold change and p value cutoffs (see below). *Puma/Bbc3* has been annotated.

Figure for referees not shown.

Figure for referees not shown.

Details for the gene ontology analysis showing effects on the p53 transcriptome are lacking. The figure legend for Supplemental Figure 4, especially parts D, E, and F need more information to clearly support the claims being made in the text.

To respond to this request from the reviewer, these figures and figure legends have been expanded and clarified in the revised manuscript.

3. 2Gy treatment is used for the RNA-seq studies whereas 5Gy is used for the tdTomato studies. The difference in radiation dose should be explained. Is Puma expression affected with 2Gy and is that found in the RNA-seq

dataset (see above)?

Our RNA sequencing dataset showed that the levels of *Puma(Bbc3)* were not different between γ -irradiated *Dnmt3a* mutant and wild type cells following 2 Gy irradiation. 5 Gy was used for the experiments using the cells from the *Puma-tdTomato* reporter mouse in this instance, because this dose was also used in the original study/paper in which these reporter mice were described (Lieschke et al. 2024). This study used a single dose of 5 Gy γ -radiation to measure reporter activity and observe a clear increase above the basal levels of the *Puma-tdTomato* reporter in the cells tested. By using 5 Gy we were able to observe the blunted expression of the *Puma-tdTomato* reporter in cells from the bone marrow or spleen from the *Dnmt3a* mutant mice, whereas other less sensitive assays were unable to capture such a difference.

4. The gene that encodes the Puma protein is called Bbc3. The author should take care to use the appropriate designation to avoid any confusion.

This has been adjusted in our revised manuscript.

Responses to Referee #2:

- **Is the effect (defects in p53-regulated pathways) related to the general observed defect in DNA methylation? DNMT3A is known to interact with p53 and control the expression of p53-dependent genes such as p21, but**

possibly in methyltransferase activity independent manner (PMID: 31640986). Paradoxically to the presented study, in some cases, DNMT3A repressed p53-dependent gene expression.

Our data confirm that the *DNMT3a*^{R878H/+} mutation causes aberrant DNA methylation, and in addition they show that this mutation causes a reduction in the levels of *p53* mRNA and a reduction in *p53* pathway activity. As the reviewer notes, *DNMT3a* can interact with *p53*, and the *Dnmt3a* R878H mutant protein may exert pro-tumorigenic effects through interaction with the *Trp53* protein. However, it remains unclear if this occurs independently of, or directly due to, defective DNA methylation.

- **Does the mutation affect p53 stability, post-translational modifications?**

Our data indicate that *DNMT3a* may regulate *p53* transcriptionally, as mutant *DNMT3A* results in reduction of *p53* mRNA as well as an overall reduction in *p53* pathway activity. If post-translational modifications of *p53* also occur, this would likely be an indirect effect of the mutation.

- **Are the observed defects in PUMA expression p53 dependent?**

No differences in the levels of *Puma/Bbc3* mRNA between γ -irradiated *Dnmt3a* mutant LSK cells vs γ -irradiated WT LSK cells were observed in our RNAseq data. However, the *Dnmt3a* R878H mutation did significantly reduce the induction of the *Puma*-tdTomato reporter in LSK cells after γ -irradiation. We believe that this method is more likely to find differences between cells of different genotypes than the RNA-seq analysis where we did not find a significant difference. It is firmly established that *Puma* is a direct *p53* target gene (Yu et al. 2001; Nakano and Vousden 2001; Han et al. 2001). We propose that the observed defects are therefore *p53* dependent. To prove this beyond doubt we would need to generate mice with the *Dnmt3a* mutation, the *Puma*-tdTomato reporter and lacking *p53* (*p53*^{-/-}). Generating and analysing would take us one year; we therefore assert that this paper should not need to wait this long for these results. We have adjusted the text in the revised manuscript accordingly.

-

- **Would the observed defects in PUMA expression be the main driver for the observed phenotypes? would potential defects in cell cycle regulating factors such as p21 be mediating the observed defects in self-renewal and/or differentiation?**

We do not wish to suggest that it is solely the reduction of *Puma* that causes the phenotype observed in *Dnmt3a* mutant mice. Rather, the abnormally increased LSK cell proliferation is likely a consequence of the reduction of *p53* levels and

activity (as a consequence of the *Dnmt3a* mutation) causing a minor reduction in many *p53* target genes, including *Puma*, rather than just one of its target genes. Possibly, a reduction in the levels and activity of *p53* will also reduce the expression of genes that encode negative regulators of cell cycling. Moreover, impact on indirect *p53* target genes could cause a decrease in genes that encode positive regulators of cell cycling. For instance, *Ccnd1* and *Ccnd2* (CyclinD1 and CyclinD2) are key regulators of cell cycle progression and survival, and both are frequently mutated in AML. The overexpression of these genes has previously been implicated in hematologic malignancy (Metcalf et al. 2010). Although not statistically significant, we found clear enrichment of mRNA of these genes in γ -irradiated *Dnmt3a* mutant LSK cells. We believe that the sum of these small effects on the expression of many genes (decrease in direct *p53* target genes and increase or decrease in indirect target genes) explains the increased proliferation of LSK cells observed in the *Dnmt3a* mutant mice. It is difficult to experimentally test this as a minor reduction in the levels of a large number of genes simultaneously is not feasible with current technology. We have expanded the discussion of these findings in our revised manuscript.

Figure for referees not shown.

Moreover, the small increase in *Mcl1* mRNA (anti-apoptotic protein), and reduced expression of DNA damage response genes (*Chek1*, *Chek2*, *Rad51*) in *Dnmt3a* mutant LSK cells, further indicate that it is the sum of many small effects that underlies the abnormally increased expansion of LSK cells in *Dnmt3a* mutant mice.

Figure for referees not shown.

The previous Figure 4 has been moved to become Expanded View Figure EV2C-D, and the above plots showing the expression levels of key genes in this gene set are now presented in Figure 5 in the revised manuscript.

- Han, J., C. Flemington, A. B. Houghton, Z. Gu, G. P. Zambetti, R. J. Lutz, L. Zhu, and T. Chittenden. 2001. 'Expression of *bbc3*, a pro-apoptotic BH3-only gene, is regulated by diverse cell death and survival signals', *Proc Natl Acad Sci U S A*, 98: 11318-23.
- Korotkevich, Gennady, Vladimir Sukhov, Nikolay Budin, Boris Shpak, Maxim N. Artyomov, and Alexey Sergushichev. 2021. 'Fast gene set enrichment analysis', *bioRxiv*: 060012.
- Labi, V., M. Erlacher, G. Krumschnabel, C. Manzl, A. Tzankov, J. Pinon, A. Egle, and A. Villunger. 2010. 'Apoptosis of leukocytes triggered by acute DNA damage promotes lymphoma formation', *Genes Dev*, 24: 1602-7.

- Lieschke, Elizabeth, Annabella Thomas, Andrew Kueh, Georgia Atkin-Smith, Pedro Baldoni, John La Marca, Savannah Young, Alan Huang, Aisling Ross, Lauren Whelan, Deeksha Kaloni, Lin Tai, Gordon Smyth, Marco Herold, Edwin Hawkins, Andreas Strasser, and Gemma Kelly. 2024. 'Mouse models to investigate in situ cell fate decisions induced by TP53 and other factors', *EMBO, in press*.
- Metcalf, R. A., S. Zhao, M. W. Anderson, Z. S. Lu, I. Galperin, R. J. Marinelli, A. M. Cherry, I. S. Lossos, and Y. Natkunam. 2010. 'Characterization of D-cyclin proteins in hematolymphoid neoplasms: lack of specificity of cyclin-D2 and D3 expression in lymphoma subtypes', *Mod Pathol*, 23: 420-33.
- Michalak, E. M., C. J. Vandenberg, A. R. Delbridge, L. Wu, C. L. Scott, J. M. Adams, and A. Strasser. 2010. 'Apoptosis-promoted tumorigenesis: gamma-irradiation-induced thymic lymphomagenesis requires Puma-driven leukocyte death', *Genes Dev*, 24: 1608-13.
- Nakano, K., and K. H. Vousden. 2001. 'PUMA, a novel proapoptotic gene, is induced by p53', *Mol Cell*, 7: 683-94.
- Weinstein, J. N., E. A. Collisson, G. B. Mills, K. R. Shaw, B. A. Ozenberger, K. Ellrott, I. Shmulevich, C. Sander, and J. M. Stuart. 2013. 'The Cancer Genome Atlas Pan-Cancer analysis project', *Nat Genet*, 45: 1113-20.
- Yu, J., L. Zhang, P. M. Hwang, K. W. Kinzler, and B. Vogelstein. 2001. 'PUMA induces the rapid apoptosis of colorectal cancer cells', *Mol Cell*, 7: 673-82.

Dear Dr. Herold,

Thank you for the submission of your revised manuscript to our editorial offices. I have already forwarded the reports from the 2 referees that I asked to re-evaluate your study, you will find again below. I also have received your provisional point-by-point-response (further revision plan). After looking through this, I decided to invite a final revised manuscript that addresses the remaining referee points as indicated in your revision plan. Please also provide a detailed final point-by-point-response to the remaining referee points.

- Please provide the abstract written in present tense throughout.

- Please order the sections like this, using these names:

Title page - Abstract - Keywords - Introduction - Results - Discussion - Methods - Data availability section - Acknowledgements - Disclosure and Competing Interests Statement - References - Figure legends - Expanded View Figure legends

- Please make sure that all the funding information is also entered into the online submission system and that it is complete and similar to the one in the acknowledgement section of the manuscript text file.

- Please use our reference format:

- Please provide individual production quality figure files as .eps, .tif, .jpg (one file per figure), of main figures and EV figures. Please upload these as separate, individual files upon re-submission.

- The nomenclature for the EV figures is not correct. Please use 'Figure EVx' instead of Expanded View Figure EVx etc. ... Please update all callouts accordingly.

- Please check again that the number "n" for how many independent experiments were performed, their nature (biological versus technical replicates), the bars and error bars (e.g. SEM, SD) and the test used to calculate p-values is indicated in the respective figure legends. Please also check that all the p-values are explained in the legend, and that these fit to those shown in the figure. Please provide statistical testing where applicable. Please avoid the phrase 'independent experiment', but clearly state if these were biological or technical replicates. Please also indicate (e.g. with n.s.) if testing was performed, but the differences are not significant. In case n=2, please show the data as separate datapoints without error bars and statistics. See also:

<http://www.embopress.org/page/journal/14693178/authorguide#statisticalanalysis>

If n<5, please show single datapoints for diagrams. Presently, many diagrams have only partial statistics or 'ns' is missing. Moreover:

- Please note that the figure 4E is mislabeled as figure 4I in the manuscript. This needs to be rectified.

- Please define the annotated p values ****/****/**/* as well as provide the exact p-values for the same in the legend of figure 3A, B, I as appropriate.

- Please indicate what */**/**** represents; if this represents p value(s) please indicate the statistical test used and where appropriate, and the exact p value in the legend(s) of figure(s) 3E, F, G, H.

- Please note that in figures 1B there is a mismatch between the annotated p values in the figure legend and the annotated p values in the figure file that should be corrected.

- Please indicate what */**/**** represents; if this represents p value(s) please indicate the exact p value in the legend(s) of figure(s) 3N-Q.

- Please note that the exact p values are not provided in the legends of figures 1B-F; 2B, G, I, K; 6E, F; EV1 B.

- Please indicate the statistical test used for data analysis in the legends of figures 4B, E; EV4 B, F, G.

- Please note that the box plots need to be defined in terms of minima, maxima, centre, bounds of box and whiskers, and percentile in the legends of figures 5B-H

- Please note that information related to n is missing in the legends of figures 2I, 3F, G-I, R; 4B, 5B-H; EV1 B-H, K; EV4 B, EV 6A

- Please note that the error bars are not defined in the legends of figures 1B-F; 2C, D, E, F, G, I, K; 3A, B, D, E, F, G-I, L-R; 6E, F; EV1 B-H, K; EV2 C, D; EV5A-D; EV6A, EV7 A.

- Please add to each legend (main, EV and Appendix figures, where applicable) a 'Data Information' section (or name the provided section like this) explaining the statistics used or providing information regarding replicates and scales. See:

- All Materials and Methods need to be described in the main text using our 'Structured Methods' format, which is required for all research articles. According to this format, the Methods section should include a Reagents and Tools Table (listing key reagents, experimental models, software, and relevant equipment and including their sources and relevant identifiers), uploaded as separate file, and a Methods section in which we encourage the authors to describe their methods using a step-by-step protocol format with bullet points, to facilitate the adoption of the methodologies across labs. More information on how to adhere to this format as well as downloadable templates (.doc) for the Reagents and Tools Table can be found in our author guidelines (section 'Structured Methods'):

- Please move the antibody information (Table EV2, Table S2) to the Reagents & Tools Table. Please update the callouts (see Reagents & Tools Table).

- Please remove the EV tables from the manuscript text file. Table 1 needs to be uploaded (once) as separate file and the correct nomenclature in all places should be Table EV1 (not Expanded View Table EV1 OR Table S1). Please do that and update all callouts.

- Please remove now the referee access information from the Data Availability section and make sure the datasets are public latest upon online publication of the paper.

- The diagrams 'DN' and 'CD8+' in panel 2K look very similar, also the grouping of the datapoints. Please check if the correct data is displayed.

In addition, I would need from you:

Best,

Referee #1:

The authors have made improvements by including additional data and clarifying their findings. They addressed concerns regarding the p53 pathway, the role of Puma, and the overall impact of the Dnmt3aR878H/+ mutation on cell proliferation and survival. The reviewer acknowledges the technical difficulties, explained by the authors, in addressing the direct link between Dnmt3a dependent effects and dysregulation of p53-dependent pathways. For example, Puma is indeed a well-established p53 dependent gene, but p53 is not the only transcription factor controlling Puma expression. This general concern, raised by both reviewers, has not been addressed. Currently the authors suggest that the observed effects are due to p53 dysregulation.

As mentioned in the 1st revision, the reviewer fully acknowledges the in depth and systematic phenotypic characterization of the DNMT3A R878H animals, which is important.

Referee #2:

Unfortunately, the authors haven't adequately addressed the concerns of the previous review. Below is cited the five points from Reviewer #1 and the continuing concern for each.

Reviewer #1 Point 1. The authors propose that alterations in the p53 transcriptome are important for the effects of the DNMT3A mutation. This needs to be tested directly by showing that the gene signature that is identified is in fact p53-dependent. Reliance on gene set enrichment is insufficient. Use of a p53-null mouse or p53 knock-down is essential to make this conclusion.

Continuing Concern:

They were specifically asked to directly show the p53-dependence of gene expression effects, either of the gene signature or Bbc3, the gene encoding Puma. They were asked not to rely solely on Gene Set Enrichment Analysis. In their response, they continue to bolster their argument solely with GSEA.

Reviewer #1 Point 2. Attention is drawn to the role of Puma in the observed outcomes. This aspect of the study needs to be substantially bolstered. Confirmation of the effects on Puma expression that are seen with the reporter need to be made by directly examining mRNA expression of Puma by RT-PCR and confirming that its transcriptional regulation is p53-dependent. The authors present data with effects of the DNMT3A mutation on global methylation levels and imply this is responsible for the effect on Puma. Methylation status on the gene encoding Puma is necessary to demonstrate this.

Continuing Concern:

mRNA for Bbc3 was checked and unlike the reporter does not change. Methylation status of the Bbc3 gene is still not shown.

Reviewer #1 Point 3. Most importantly, the Puma-dependence of the effects of the DNMT3A mutation need to be directly demonstrated. The reporter strain that is used has a knock-in of tdTomato making the Puma allele effectively null. The authors should use this mouse strain in the homozygous state to show that effects on thymic lymphoma development and LSK cell function are indeed Puma-dependent.

Continuing Concern:

They have not done do these experiments, arguing they would be uninformative. Further, they argue that they do not mean to ascribe the observed effects solely to Puma. But without that connection, the manuscript lacks mechanistic insight.

Reviewer #1 Point 4. Puma has a well-characterized role in apoptosis. The observed phenotype appears to be associated with enhanced proliferation of LSK cells. The authors need to address how reduced expression of Puma contributes to this behavior of the LSK cells. Figure 4 is uninformative as there appears to be no statistical differences between wild-type and mutant.

Continuing Concern:

They again argue that they do not wish to ascribe this effect to Puma. Rather it is the accumulation of small effects on gene expression, many without statistical significance. This again does not give mechanistic insight.

Reviewer #1 Point 5. The authors should also discuss further the gene set that is considered to be p53-dependent. What other well-known p53 targets besides Puma were found? Can the regulation of a subset of these be confirmed by RT- qPCR? And as noted above, is their expression indeed p53-dependent?

Continuing Concern:

The authors make the same GSEA-based argument as in their response to Point 1. This is unacceptable.

Re: Submission of Revised Manuscript ID: EMBOR-2024-60834

Point-by-point response to the referees' second round of concerns

Referee #1:

The authors have made improvements by including additional data and clarifying their findings. They addressed concerns regarding the p53 pathway, the role of Puma, and the overall impact of the Dnmt3aR878H/+ mutation on cell proliferation and survival. The reviewer acknowledges the technical difficulties, explained by the authors, in addressing the direct link between Dnmt3a dependent effects and dysregulation of p53-dependent pathways. For example, Puma is indeed a well-established p53 dependent gene, but p53 is not the only transcription factor controlling Puma expression. This general concern, raised by both reviewers, has not been addressed. Currently the authors suggest that the observed effects are due to p53 dysregulation.

As mentioned in the 1st revision, the reviewer fully acknowledges the in depth and systematic phenotypic characterization of the DNMT3A R878H animals, which is important.

We thank Referee #1 for their highly complimentary remarks about our work, their feedback, and their understanding regarding the technical challenges making it difficult to undertake further experiments. The referee notes a general concern that Puma/Bbc3 can also be regulated by transcription factors other than p53. While this is of course true, evidence indicates that p53 is the primary driver of Puma/Bbc3 induction in response to DNA damage, particularly following γ -irradiation [1, 2]. While Puma can certainly be regulated by other transcription factors in response to different apoptotic stimuli, such as treatment with glucocorticoids, in this scenario of γ -irradiation induced DNA damage, p53 is the predominant if not sole driver of induction of *Puma/Bbc3* transcription. This has been demonstrated in studies showing that Puma is induced in a p53-dependent manner following γ -irradiation in many tissues and cell types, including lymphoid as well as other haemtopoietic cells, MEFs, neurons and intestinal crypts [3]. Our observed reduction in *Puma* transcriptional reporter activity in Dnmt3a mutant cells therefore strongly aligns with the notion of attenuated p53 activity following γ -irradiation in this context. We have updated the text with a sentence and citation (See Text line 323-328 in the Track changes manuscript).

Referee #2:

#1 Continuing Concern:

They were specifically asked to directly show the p53-dependence of gene expression effects, either of the gene signature or Bbc3, the gene encoding Puma. They were asked not to rely solely on Gene Set Enrichment Analysis. In their response, they continue to bolster their argument solely with GSEA.

We thank Referee #2 for their feedback, and for the opportunity to address their concerns. In our previous response to this concern, in addition to the GSEA, we included data from *Dnmt3a*^{R878H/+}/*Trp53*^{+/-} mice. We showed that mutant Dnmt3a does not cooperate with loss of one allele of *Trp53* to accelerate tumour development. Of note, this result is consistent with existing datasets, including in the human AML context, where mutations in the genes encoding DNMT3A and P53 tend not to co-occur. We proposed that this is because Dnmt3a^{R878H/+} causes deregulation of the p53 pathway to the threshold needed for driving tumorigenesis. The additional removal of one allele of *Trp53* in this context therefore does not produce additive effects in accelerating disease. We explained also that while we attempted to generate a cross between Dnmt3a^{R878H/+} mice and homozygous p53-null mice, this was not successful due to breeding difficulties. Furthermore, in addition to the GSEA analysis, we investigated the mRNA levels of individual genes in the p53 pathway. We acknowledged fully that none of the differences in individual gene expression were statistically significant but noted that the consistent pattern of subtle downregulation of many p53 pathway genes was still likely to be biologically relevant, and indicative of mutant DNMT3a causing a reduction in overall p53 pathway signaling. We appreciate that Referee #1 acknowledged that this concern was addressed to their satisfaction.

#2 Continuing Concern:

**mRNA for Bbc3 was checked and unlike the reporter does not change.
Methylation status of the Bbc3 gene is still not shown.**

The PumatdTomato reporter was intended to be a more sensitive assay to identify changes in Puma/Bbc3 (and indirectly Trp53) activity in the Dnmt3a^{R878H/+} mutant context, for which transcriptomic analysis did not identify statistically significant differences. The reason for this are that the effects of mutant Dnmt3a on p53 pathway activity are subtle but highly consistent across many p53 target genes. We have shown that Dnmt3a^{R878H/+} has a global hypomethylation phenotype. Hypomethylation of genes such as *Trp53* and others could each contribute to functional impairments in *Puma/Bbc3* transcription, even if there is no differential methylation surrounding the *Puma/Bbc3* gene itself. We have made it clearer in the text that the Puma reporter was used as a more sensitive assay to identify changes in Puma/Bbc3 (and indirectly p53) activity in the Dnmt3a^{R878H/+} mutant context, for which transcriptomic analysis did not identify statistically significant differences. See *updated Text in Track Changes manuscript lines 346 -352*.

#3 Continuing Concern:

They have not done do these experiments (using homozygous Puma reporters to show the effect on thymic lymphoma development), arguing they would be uninformative. Further, they argue that they do not mean to ascribe the observed effects solely to Puma. But without that connection, the manuscript lacks mechanistic insight.

In our previous response, we pointed out that homozygous Puma reporters cannot be used to observe the effect on the development of thymic T cell lymphoma, as the absence of Puma (which occurs with homozygosity of the Puma-tdTomato reporter allele >> see E Lieschke, EMBO J 2024) prevents γ -radiation induced thymic T cell lymphoma development (EM Michalak et al, Genes & Development 2010; V Labi et al, Genes & Development 2010). Respectfully, our argument was not that the referee's proposed experiment would be uninformative, but that it would not be possible to test in this context.

#4 Continuing Concern:

They again argue that they do not wish to ascribe this effect to Puma. Rather it is the accumulation of small effects on gene expression, many without statistical significance. This again does not give mechanistic insight.

We appreciate the referee's comment and understand the desire for deeper mechanistic insights. In our previous response, we acknowledged that although not statistically significant, there is an accumulation of subtle gene expression changes that we propose cumulatively contribute to the phenotypes observed in the Dnmt3a^{R878H/+} mutant mice. While our study does not delineate the precise mechanism linking the Dnmt3a^{R878H/+} mutation to this downregulation of the expression of several genes, the existing literature supports our interpretation. It has been established that p53 protein levels are central to governing the hierarchy of DNA damage response pathways, and even modest changes in expression levels of multiple p53 target genes can influence cellular outcomes [4]. In addition to p53 protein level dynamics, other aspects such as post-translational modifications (including methylation) of p53 and its special location can play crucial roles in controlling cellular responses to DNA damage [5]. Our observed reduction in Puma-tdTomato reporter induction after γ -radiation in Dnmt3a^{R878H/+} mutant cells provides functional evidence consistent with attenuated p53 activity resulting from this mutation in DNMT3a. While more detailed mechanistic insights remain to be elucidated, our findings provide a strong foundation for future studies investigating the contribution of abnormalities in the expression of p53 regulated genes to the disease states caused by the Dnmt3a^{R878H/+} mutation.

#5 Continuing Concern:

The authors make the same GSEA-based argument as in their response to Point 1. This is unacceptable.

The referee's original request was a further discussion of the gene set, and to identify p53 target genes besides *Puma/Bbc3* which were found in our analysis. Therefore, we expanded our transcriptomic analysis and included additional known p53 target genes

which were deregulated in Dnmt3a mutant cells in our manuscript. We note that currently available technology does not make it possible to experimentally test the impact of the minor reduction in the expression levels of several genes simultaneously.

1. Kuchur, O.A., et al., *Differential Regulation of BBC3/PUMA and PMAIP1/Noxa in Ionizing Radiation: the Role of p53*. *Cell and Tissue Biology*, 2021. **15**(6): p. 544-553.
2. Jeffers, J.R., et al., *Puma is an essential mediator of p53-dependent and -independent apoptotic pathways*. *Cancer Cell*, 2003. **4**(4): p. 321-8.
3. Yu, J. and L. Zhang, *PUMA, a potent killer with or without p53*. *Oncogene*, 2008. **27 Suppl 1**(Suppl 1): p. S71-83.
4. Castaño, B.A., et al., *The levels of p53 govern the hierarchy of DNA damage tolerance pathway usage*. *Nucleic Acids Res*, 2024. **52**(7): p. 3740-3760.
5. Luo, Q., et al., *Dynamics of p53: A Master Decider of Cell Fate*. *Genes (Basel)*, 2017. **8**(2).

Dr. Marco Herold
The Walter and Eliza Hall Institute
Blood Cells and Blood Cancer
1 G Royal Parade
Parkville
Melbourne 3052
Australia

Dear Dr. Herold,

Thank you for the submission of your further revised manuscript to our editorial offices. I have now looked through the files and your final p-b-p-response, and consider the remaining referee concerns and the editorial requests as adequately addressed.

I am thus very pleased to accept your manuscript for publication in the next available issue of EMBO reports. Thank you for your contribution to our journal.

Yours sincerely,
